# Multi-label classification of symptom terms from free-text bilingual adverse drug reaction reports using natural language processing

**Sitthichok Chaichulee**[1,5]*, **Chissanupong Promchai**[2], **Tanyamai Kaewkomon**[2], **Chanon Kongkamol**[3,5], **Thammasin Ingviya**[3,5], **Pasuree Sangsupawanich**[4]

**1** Department of Biomedical Sciences and Biomedical Engineering, Faculty of Medicine, Prince of Songkla University, Hatyai, Songkhla, Thailand, **2** Department of Pharmacy, Songklanagarind Hospital, Faculty of Medicine, Prince of Songkla University, Hatyai, Songkhla, Thailand, **3** Department of Family and Preventive Medicine, Faculty of Medicine, Prince of Songkla University, Hatyai, Songkhla, Thailand, **4** Department of Pediatrics, Faculty of Medicine, Prince of Songkla University, Hatyai, Songkhla, Thailand, **5** Division of Digital Innovation and Data Analytics, Faculty of Medicine, Prince of Songkla University, Hatyai, Songkhla, Thailand

* sitthichok.c@psu.ac.th

**Data Availability Statement:** We can not published our datasets along with the manuscript because of patient confidentiality. Our unstructured data

## Abstract

Allergic reactions to medication range from mild to severe or even life-threatening. Proper documentation of patient allergy information is critical for safe prescription, avoiding drug interactions, and reducing healthcare costs. Allergy information is regularly obtained during the medical interview, but is often poorly documented in electronic health records (EHRs). While many EHRs allow for structured adverse drug reaction (ADR) reporting, a free-text entry is still common. The resulting information is neither interoperable nor easily reusable for other applications, such as clinical decision support systems and prescription alerts. Current approaches require pharmacists to review and code ADRs documented by healthcare professionals. Recently, the effectiveness of machine algorithms in natural language processing (NLP) has been widely demonstrated. Our study aims to develop and evaluate different NLP algorithms that can encode unstructured ADRs stored in EHRs into institutional symptom terms. Our dataset consists of 79,712 pharmacist-reviewed drug allergy records. We evaluated three NLP techniques: Naive Bayes—Support Vector Machine (NB-SVM), Universal Language Model Fine-tuning (ULMFiT), and Bidirectional Encoder Representations from Transformers (BERT). We tested different general-domain pre-trained BERT models, including mBERT, XLM-RoBERTa, and WanchanBERTa, as well as our domain-specific AllergyRoBERTa, which was pre-trained from scratch on our corpus. Overall, BERT models had the highest performance. NB-SVM outperformed ULMFiT and BERT for several symptom terms that are not frequently coded. The ensemble model achieved an exact match ratio of 95.33%, a $F_1$ score of 98.88%, and a mean average precision of 97.07% for the 36 most frequently coded symptom terms. The model was then further developed into a symptom term suggestion system and achieved a Krippendorff's alpha agreement coefficient of 0.7081 in prospective testing with pharmacists. Some degree of automation could both accelerate the availability of allergy information and reduce the efforts for human coding.

records may contain patients' personal information. Data are however available from the corresponding authors or the Division of Digital Innovation and Data Analytics upon request and with permission from the Office of Human Research Ethics Committee (HREC), Faculty of Medicine, Prince of Songkla University. The Division of Digital Innovation and Data Analytics oversees the use of institutional clinical data for research and can be contacted at dida@psu.ac.th.

**Funding:** The authors received no funding for this work.

**Competing interests:** The authors have declared that no competing interests exist.

## Introduction

All medications can cause non-allergic side effects and allergies. Drug allergy occurs when the immune system, which is supposed to protect the body from disease, reacts to the drug being taken, resulting in increased inflammation [1]. Drug allergy can negatively impact a person's quality of life and lead to treatment delays and even death.

Drug allergy involves a wide range of immunologically-mediated hypersensitivity reactions with different mechanisms and clinical symptoms [2, 3]. Drug-induced allergic responses can impact a variety of organ systems and cause a variety of symptoms. Common symptoms caused by drug allergy include rash, edema, fever, diarrhea, wheezing, even shortness of breath. Skin manifestations are the most common sign of drug-induced allergic reactions. The most severe skin reactions include Stevens-Johnson Syndrome (SJS) and Toxic Epidermal Necrolysis (TEN) [4]. SJS is a serious disorder of the skin with erythema, purpura, and blisters, whereas TEN causes widespread erythema, necrosis, and skin detachment. Many drug-induced reactions might have more than one mechanism. Other mechanisms may include hematologic problems, swollen lymph nodes, and liver inflammation. Patients who have multiple drug allergies may actually suffer from underlying chronic diseases [1]. Anaphylaxis is a severe systemic hypersensitivity reaction that occurs as a result of multi-organ responses to medication.

Medical attention is needed immediately if an allergic reaction is suspected. Different medications have different effects. Certain medications may cause more allergic reactions in people than other medications. Drug-induced allergic reactions can vary greatly. Clinical conditions that resemble drug-induced allergic reactions must be ruled out. It is important to document drug allergy at any stage of care. A detailed review of patient medical records is crucial for evaluating patients with potential drug allergy. The most effective approach for managing drug allergy is to avoid or stop any drug in question entirely [3]. Substitute medications with unrelated chemical structures should be used. If patients have mild symptoms, they may be prescribed with additional medications to control drug reactions.

### Adverse drug reactions

Obtaining allergy information from patients is an important step toward safe prescription, prevention of adverse drug events, and reducing the cost of care. Clinical staff routinely asks for the history of adverse drug reactions (ADRs) during a patient interview and records into the electronic health record (EHR); however, the documentation of drug allergy is often incomplete. ADRs are often recorded in an unstructured free-text format, even the system provides with an ability to document a report in a coded manner using a drop-down selection with a list of specific medications, reactions, or hypersensitivities. Clinical staff often also include supplementary information beyond the specific allergy history in the free-text box, especially if the allergy is uncertain. Pharmacists or allergists have to later manually review and code each allergy record to ensure proper documentation.

Maintaining a list of a patient's active ADRs in a coded manner is critical to patient safety. It allows for the development of an automated system to alert against prescribing medications that may cause allergies, hypersensitivities, or drug adverse events. If ADRs have not been coded correctly, physicians, and pharmacists will need to review all free-text records prior to prescribing any medication to ensure patient safety. Exchange of an active ADRs between different healthcare systems using the same standard could also alert other healthcare providers when attempting to prescribe drugs to which the patient is known to be allergic [2]. More than 10% of medication prescribing errors are related to the failure to recognize a patient's previously known drug allergy [5].

## Natural language processing

The widespread adoption of EHRs means that a large amount of clinical data is generated every day, opening new avenues for clinical research [6]. The role of machine learning (ML) as a tool to improve patient care is becoming increasingly important. Natural language processing (NLP) is a branch of computer science and computational linguistics that enables interaction between computers and humans. Currently, unstructured clinical data reportedly accounts for more than 80% of all currently available health data [7]. NLP could enable automated analysis of clinical text to identify relevant information and reduce methodological differences in phenotyping clinical data.

NLP enables the transformation of unstructured human language into structured data [8]. Its approaches are increasingly used to extract important information from EHRs. They can be used to extract key concepts or clinical phenotypes from clinical texts. NLP differs from a simple keyword search in that it can learn to distinguish different notes from different clinicians. An NLP algorithm is trained with a training cohort. Once its optimal performance is achieved, the NLP algorithm is evaluated with an independent test cohort. NLP can be used for a variety of purposes, including clinical care and research studies, as well as hospital quality improvement or other population-based purposes.

For many years, text classification was achieved by using traditional rule-based or statistical approaches [8]. Words are represented as one-dimensional vectors with one-hot encoded values. This representation results in long sparse vectors for each sentence that may not reflect the context of the word. One of the most effective techniques is Naive Bayes—Support Vector Machine (NB-SVM) [9], which combines Bayesian probabilities involving word counts with a linear SVM model for classification. Although traditional text classification methods have been shown to be successful and fast, they do not understand the context of words and their development requires a large manually annotated dataset for each specific application.

In NLP, a language model (LM) is used for predicting the probability of occurrence of different linguistic units (e.g. words, sentences, or paragraphs) occurring in a sequence [10]. LM works like a human learning process, not only in word prediction but also in language comprehension. It has been shown that the hypothesis space generated by LM is useful for a variety of different NLP tasks. LM is usually pre-trained on a large corpus of millions of documents containing over a hundred million words to capture generic language characteristics (the self-supervised training of LM is called pre-training) [10, 11]. The pre-training phase is the most expensive, but it only needs to be done once. After LM is trained, it can be fine-tuned to downstream tasks. Recently, deep learning (DL) has brought the new era of language models that are developed based on a huge text corpus of a billion or trillion words with an enormous learning capacity over billions of parameters. LMs have shown promise in data retrieval, text classification, text summarization, and sentiment analysis.

Universal Language Model Fine-tuning (ULMFiT) is an approach for fine-tuning LMs to downstream tasks [10]. ULMFiT addresses the problems of fine-tuning LMs on small datasets as well as catastrophic forgetting, which happens when LMs lose previously learned information while learning new information. ULMFiT utilizes a shallow Long Short-Term Memory (LSTM) network, enabling the learning of analyze long-distance word dependencies in a sentence [8]. LSTM normally takes the previous state and the current input and determines which to maintain and which to discard. In ULMFiT, a LM is first pre-trained on a very large general-domain corpus and later fine-tuned on a specific-domain target corpus. The LM is then augmented with a classifier and fine-tuned for a target task. Instead of aggressively fine-tuning all weight layers, ULMFiT gradually fine-tunes each layer of the LM starting from the last layer which holds the least general knowledge, allowing stable and successful learning [10].

The mechanism of LSTM-based LMs, which processes each word sequentially, can cause performance degradation for long sentences or paragraphs. To address this problem, a self-attention mechanism was developed to associate each word in the sentence with the word in focus [11]. Transformer-based LMs consist of an encoder, which maps a sequence into internal state vectors, and a decoder, which maps the encapsulated vectors back into the sequence. Transformer-based LMs can be coupled with bidirectional processing and positional encoding to form Bidirectional Encoder Representations from Transformers (BERT) [11] which can mimic certain characteristics of continuous learning in humans. This approach can provide improved capabilities for dealing with unstructured text in a variety of real-world applications. Recently, transformer-based LMs have become the de facto standard in NLP applications.

There are many public BERT-based models that have been pre-trained on general-domain and domain-specific data. Some models have been pre-trained to learn multilingual language representation, such as XLM-RoBERTa [12]. Several LMs have been pre-trained on biomedical and clinical corpora. BioBERT was pre-trained over 18 billion words from PubMed abstracts and PubMed Central full-text articles [13] ClinicalBERT was pre-trained on 2 million de-identified clinical notes from the Beth Israel Deaconess Medical Center [14]. BioALBERT [15] was pre-trained on the datasets used by BioBERT and ClinicalBERT, so the model learns language representation from both biomedical and clinical domain. All three models were reported to outperform the state-of-the-art models pre-trained on general-domain data on most downstream tasks. However, they were all pre-trained with a monolingual English corpus. For the use of BERT models on clinical data, it was recommended to pre-train the model on a private dataset at the practitioner's institution for best results, as clinical notes may vary depending on the clinical setting [14].

## Gaps and objectives

ADRs contain data that is useful for patient safety and clinical research. Although the records are part of EHRs, they are sometimes documented in an unstructured free-text format that is difficult for computers to interpret and categorize. Pharmacists or allergists may need to manually review the unstructured information and manually code it into the structured content. Accurate documentation of the patient's symptoms allows the exact physiology associated with the symptoms to be brought up. This allows for effective management of drug allergy in patients. This could also guide the way through the accurate allergy pattern, especially in patients with a multitude of drug allergy.

The processing of free-text allergy entries is, however, inherently a complex task. It is subject not only to computational challenges related to NLP in general (e.g. word tokenization, abbreviation, misspelling, implicit information, multilingualism, and ambiguity), but also to the variability in how clinical information are recorded by different healthcare professionals. An ADR often contains multiple data elements such as drug name, symptoms, severity, and episode. Encoding these data elements in standard terminologies requires knowledge for mapping the semantic and syntactic relationships between different data elements. Recently, there have been a few studies on the topics [5, 16–20]. Wagholikar *et al.* [16] developed a rule-based method using the SNOMED-CT (Systematized Nomenclature of Medicine – Clinical Terms) ontology for categorizing free-text chief complaints into symptom groups. The development of such algorithms was challenging due the different ways in which medical narratives were recorded by clinical staff. To deal with variations in clinical notes, Epstein *et al.* [5] developed an allergy-matching method that incorporates RxNorm, abbreviation, and misspelling terms to identify medication and food allergies from unstructured allergy records. Despite the use of

extensive look-up tables, some free-text entries in the test set were not matched by the algorithm. Goss *et al.* [17] conducted a similar study with a larger set of terms, including SNOMED-CT and ICD-10 (International Classification of Diseases – 10th Revision) terms. Due to the presence of non-standard terms in free-text entries, algorithmic performances still varied. Similarly, Jackson *et al.* [18] developed SVM models to specifically capture key symptoms of severe mental illness from free-text discharge summaries using disease symptomatology concepts. Some symptoms were missed due to non-standard language usage. In languages other than English, Lenivtceva *et al.* [19] developed an extensive rule-based model for coding Russian free-text medical records into allergy identification categories that complies the HL7 FHIR (Health Level 7 – Fast Healthcare Interoperability Resources) standard. The method still suffered from lack of generalizability due to the availability of data dictionaries in Russian. Recently, Leiter *et al.* [20] using deep learning to identify symptoms in patients with congestive heart failure to assess the response of cardiac resynchronization therapy using an expert-labeled set of free-text notes without the representation of medical concepts. The algorithm was, however, inconsistently performed with typographical errors and phrasings that were absent from the training set. The problem could alleviate with a larger set of training samples. Some of the studies presented thus far required the modeling of medical concepts or the extensive use of data dictionaries. Most of the studies were conducted in a monolingual setting. However, in settings where English is not the official language, EHRs are often documented in another language or even a mixture of multiple languages.

In this study, we examine the feasibility of mapping the unstructured free-text bilingual allergy description entered by clinical staff to the established symptom terms using NLP approaches on an institutional dataset. One of our main challenges is that ADRs are documented using a mixture of Thai and English words. The algorithms, hence, need to be capable of representing drug allergy information in both languages. We examined different NLP algorithms (NB-SVM, ULMFiT, and BERT) for coding the unstructured allergy description to the institutional symptom terms. We tested different general-domain multilingual pre-trained BERT models, including mBERT, XLM-RoBERTa, and WangchanBERTa, as well as our domain-specific AllergyRoBERTa, which was pre-trained from scratch on our corpus. The algorithms were trained using pharmacist-verified ADRs from Songklanagarind Hospital in Thailand over the past 18 years. We then developed the best-performing model into a symptom term suggestion system and prospectively tested against the coding of the symptom terms by expert pharmacists. Our contributions were on the investigation on the applicability of recent NLP approaches in a domain-specific clinical context, i.e. drug allergy, where there are still gaps in the development of such algorithms, especially in a setting where clinical documentation is multilingual.

Our study suggests opportunities to enhance EHRs to perform real-time mapping for ADR reporting. As clinical staff type into the free-text box, relevant symptom terms could be suggested from a list of institutional symptom terms. This allows more accurate reporting and reduces the burden of pharmacists and allergists for reviewing and verification.

## Materials and methods

This section describes the dataset used in the study, the methodology for developing algorithms, and the performance metrics for evaluating the algorithms.

Fig 1 shows a diagram illustrates our study flowchart. First, drug allergy records that were reviewed and validated by pharmacists were collected from the database of the Songklanagarind Hospital's EHR. Second, multi-label text classification algorithms were developed to extract symptom terms from free-text description. Finally, the ensemble model was further

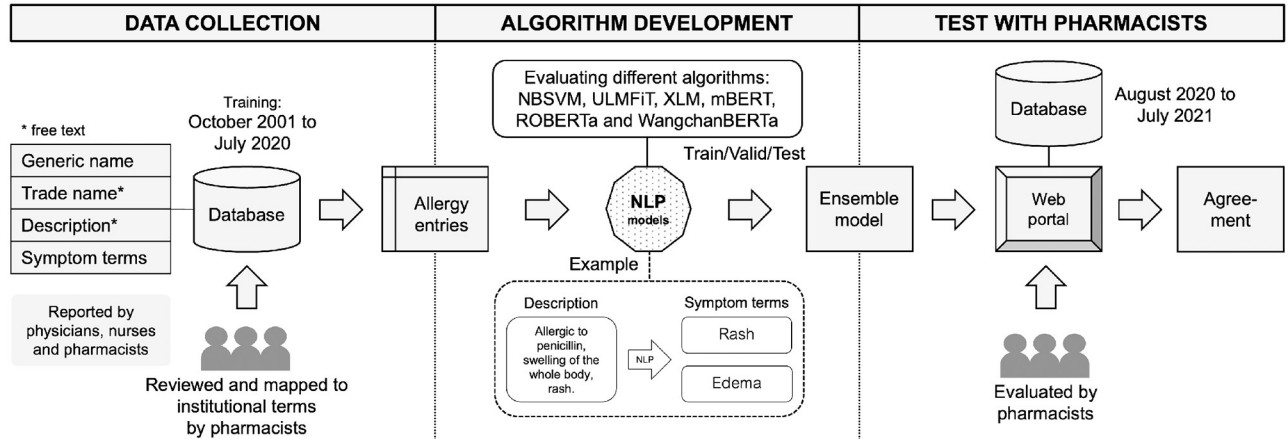

**Fig 1. Flowchart outlines the steps involved in our study.** Drug allergy records in the Songklanagarind Hospital's EHR from October 2001 to July 2020 were extracted for developing algorithms. Only the records that had been reviewed by pharmacists were considered. We then trained NB-SVM, ULMFiT, and BERT-based models to map the unstructured allergy description to the institutional symptom terms. The ensemble model was then used for evaluation with pharmacists via a web application in a simulated EHR environment.

developed into a web application that can suggest symptom terms based on free-text descriptions to assess the agreement with pharmacists.

## Data collection

Drug allergy data were collected from the ADR reporting system of the Songklanagarind Hospital's EHR. This study used anonymized patient data and was approved by the Human Research Ethics Committee (HREC), Faculty of Medicine, Prince of Songkla University (REC. 64–100-25-2) with a waiver for written informed consent due to its retrospective nature. Our institution has not implemented hospital information exchange, so the data were only from our institution.

At Songklanagarind Hospital, drug allergy can be reported by physicians, nurses, and pharmacists at any stage of care. Fig 2 illustrates a mock graphical user interface (GUI) for documenting a new drug allergy record in the Songklanagarind Hospital's EHR. The report includes a generic name (the name of active ingredients, e.g. "paracetamol"), free-text trade name (the name given by the manufacturing company, e.g. "CEMOL paracetamol syr 120mg/ 5mL"), free-text description, and multiple-choice symptom terms. A list of symptom terms was determined by a panel of pharmacists, allergists, and physicians in our institution. Each record can have zero or more than one symptom terms selected. The required-entry fields are the generic name and description. Drug allergy can be reported by physicians, nurses, and pharmacists at any stage of care. Once reported, each drug allergy entry is reviewed by pharmacists who can modify the entry, code it in the structured format, and perform further investigation, if necessary. Hence, our gold standard labels are the symptom terms that are reviewed by pharmacists.

We included the drug allergy records between October 2001 and July 2020 (18 years and 10 months) to develop our algorithms. We excluded food allergy records and drug allergy records without any symptom terms assigned. Our dataset contains 79,912 drug allergy records. Symptom terms with a prevalence of less than 0.01% were excluded. Our dataset then has 36 most frequently coded symptom terms. We have over 147,349 symptom terms assigned for all drug allergy records. Note that underreporting of adverse drug reactions is common because most

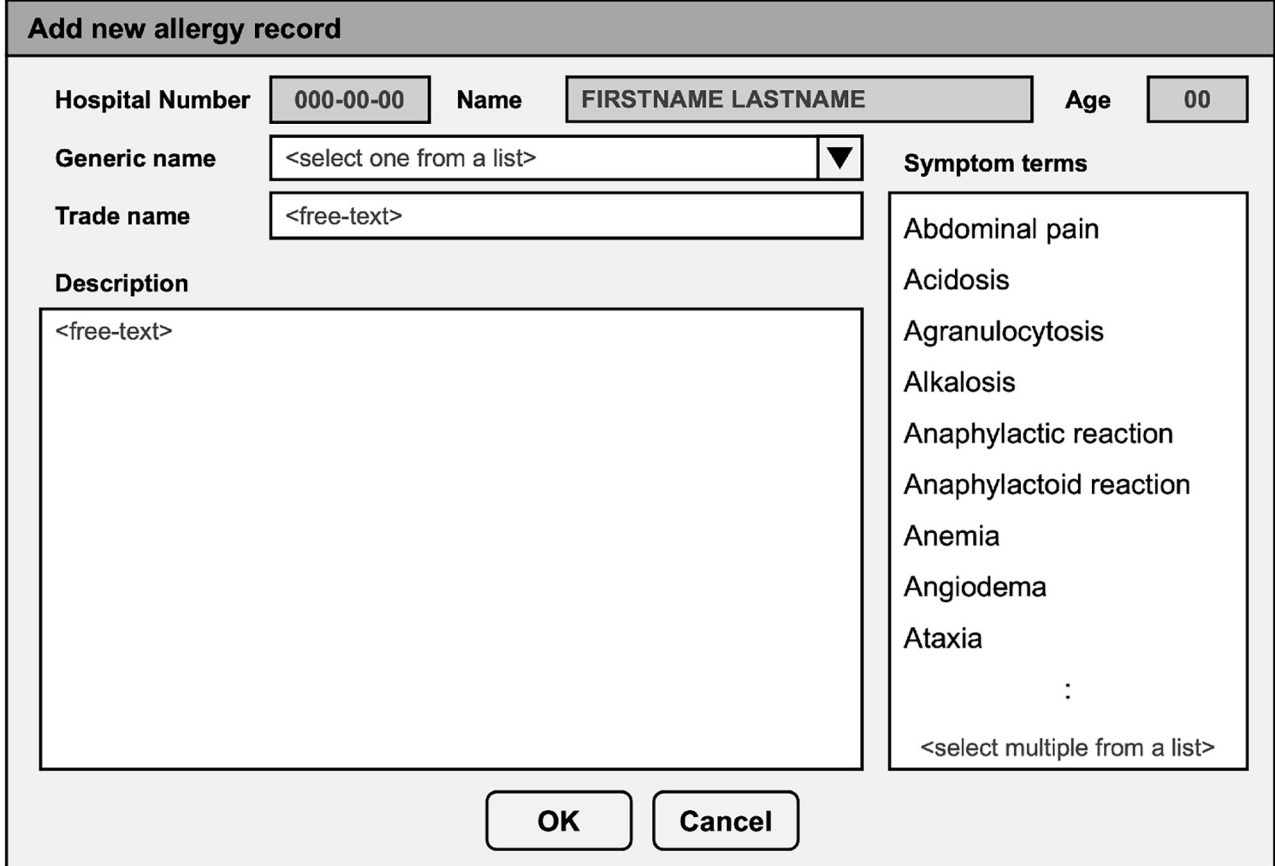

**Fig 2. GUI for document a new drug allergy record.** GUI for documenting a drug allergy record in the Songklanagarind Hospital's EHR. The user interface was originally in Thai and was translated to English for ease of the reader.

countries primarily rely on the spontaneous or voluntary method of reporting drug allergies [21].

## Data exploration

Table 1 shows the descriptive statistics of our dataset. We have 79,912 drug allergy records from 58,524 patients with an average of 1.4 records per patient in our dataset. There were 147,295 symptom terms assigned to all drug allergy records with an average of 1.8 symptom terms per record. Each drug allergy entry contains both Thai and English words. Thai words are generally used to describe relevant symptoms and other supplementary details, while English words are used to indicate drug names or symptom terms that cannot be described in Thai. Most allergy descriptions were mixed in both languages. Each drug allergy description consisted of an average of 19.7 words per record with a large standard deviation of 43.7. On average, there were 8.0 English words and 11.7 Thai words per record.

Table 2 shows the number of symptoms in each symptom term. Our dataset is highly unbalanced with the frequency of each category ranges from 20.79% (for rash) to 0.01% (for hypertension and hemolysis).

**Table 1. Dataset characteristics.**

|  | N |
| --- | --- |
| Number of records | 79,912 |
| Number of patients | 58,524 |
| Number of records per patient | 1.4 |
| Number of symptom terms | 36 |
| Number of symptom terms assigned for all records | 147,295 |
| Number of symptom terms assigned per record | 1.8 ± 1.2 |
| Number of words per record# | 19.7 ± 43.7 |
| Number of English words per record# | 8.0 ± 24.1 |
| Number of Thai words per record# | 11.7 ± 22.5 |
| Number of characters per English word# | 3.2 ± 3.6 |
| Number of characters per Thai word# | 3.9 ± 1.9 |

# Values shown in mean ± standard deviation

## Data preparation

Free-text descriptions of drug allergy were a mixture of Thai and English words, with Thai words often used to describe patient symptoms and English words often used to describe medications (e.g., aspirin) or symptom terms (e.g., edema). Unlike English, the Thai language has its own unique alphabet that includes 72 characters: 44 consonants and 28 vowels. Each word is formed by a string of characters. The Thai language is written from left to right and without punctuation or spaces between words. Word segmentation was then applied to the allergy description based on the bidirectional maximum matching algorithm on the Thai corpus [22] to separate each word with a space character. All new line characters and punctuation were replaced with spaces.

The dataset was divided into training, validation and test sets with a ratio of 0.8 (63,936 samples), 0.1 (7,988 samples), and 0.1 (7,988 samples), respectively, using stratified random sampling to ensure the balance of symptom terms in all the sets. The training set was used to train the models, while the validation set was used to determine the best configurations of the model hyperparameters. The test set was used to evaluate the performance of the models.

## Model development

Our problem can be formulated as a multi-label text classification problem, in which an algorithm processes a free-text drug allergy description and gives a score on whether the description belongs to each symptom term. Our goal is to train a model that can understand bilingual word representations of Thai and English. We evaluated three NLP algorithms for our task: NB-SVM, ULMFiT, and BERT. Each method has its own advantages and disadvantages. NB-SVM is based on word frequencies and a strong discriminative classifier. ULMFiT is an technique to fine-tune a pre-trained sequential LSTM-based LM to a target corpus and then fine-tune a classifier to a target task. BERT is an approach for fine-tuning the bidirectional attention-based LM that has been pre-trained on a very large text corpus to a target task. The architecture of BERT-based LM is more complex and larger than that of LSTM-based LM. Fig 3 outlines the diagrams of each step involved for each algorithm.

**Naive Bayes—Support Vector Machine (NB-SVM).** A Support Vector Machine model (SVM) with Naïve Bayes (NB) features is often used as a baseline for text classification [9].

**Table 2. Number of occurrences for each symptom term grouped according to organ systems.**

| Symptom term | N |
| --- | --- |
| **Anaphylaxis A** | |
| Anaphylactic reaction | 40 |
| **Anaphylaxis B** | |
| Anaphylactoid reaction | 829 |
| Anaphylaxis | 853 |
| Shock | 416 |
| **Cardiovascular system** | |
| Edema | 6,795 |
| Hypertension | 17 |
| Hypotension | 194 |
| **Nervous system** | |
| Insomnia | 35 |
| **Integumentary system** | |
| Angioedema | 22,762 |
| Eczema | 286 |
| Erythema multiforme | 717 |
| Erythematous rash | 10,543 |
| Maculopapular rash | 9,817 |
| Papular rash | 7,232 |
| Pruritus | 20,467 |
| Purpura | 118 |
| Rash | 30,616 |
| SJS/TEN | 745 |
| Urticaria | 5,952 |
| **Endocrine system** | |
| Acidosis | 41 |
| **Digestive system** | |
| Abdominal pain | 518 |
| Diarrhea | 421 |
| Nausea | 7,072 |
| Vomiting | 3,919 |
| **Hematopoietic system** | |
| Agranulocytosis | 49 |
| Anemia | 52 |
| Hemolysis | 17 |
| Neutropenia | 39 |
| Pancytopenia | 26 |
| Thrombocytopenia | 26 |
| **Hepatic portal system** | |
| Hepatitis | 327 |
| Jaundice | 77 |
| **Respiratory system** | |
| Cough | 2,081 |
| Dyspnea | 10,154 |
| Respiratory depression | 2,461 |
| **Miscellaneous** | |
| Fever | 1,581 |

(*Continued*)

**Table 2.** (Continued)

| Symptom term | N |
|---|---|
| Total | 147,295 |

\* We excluded allergy records with no symptoms reported. We also excluded symptom terms with a prevalence of less than 0.01%. Our dataset contains 79,912 allergy records with a total of 147,295 symptom terms reported from October 2001 to July 2020. Each allergy record can be associated with one or more symptom terms. One patient can have multiple allergy records.

NB-SVM combines the strong discriminative power of a linear SVM with the NB features. NB-SVM is described in detail in S1 Appendix.

For our multi-class multi-label classification problem, we used the one-vs-rest strategy, a heuristic method for using multiple binary classifiers for multi-class classification. The documents are first vectorized into a document-term matrix with unigrams and bigrams using the vocabulary list derived from all possible words in the entire document set. The matrix represents the frequency of each word in each document. For each label, Naive Bayes log-count ratios were computed to capture the probability of a word appearing in a document in one label compared to the other labels. A separate binary SVM model was then learned for each label. These resulted 36 binary NB-SVM classifiers. We addressed the data imbalance problem with the class weighting scheme. The C parameter was weighted by the size of the classes, so we penalized SVM more harshly for misclassifying less frequent classes than for misclassifying the frequent classes. For inference, a document was converted into a document-term matrix

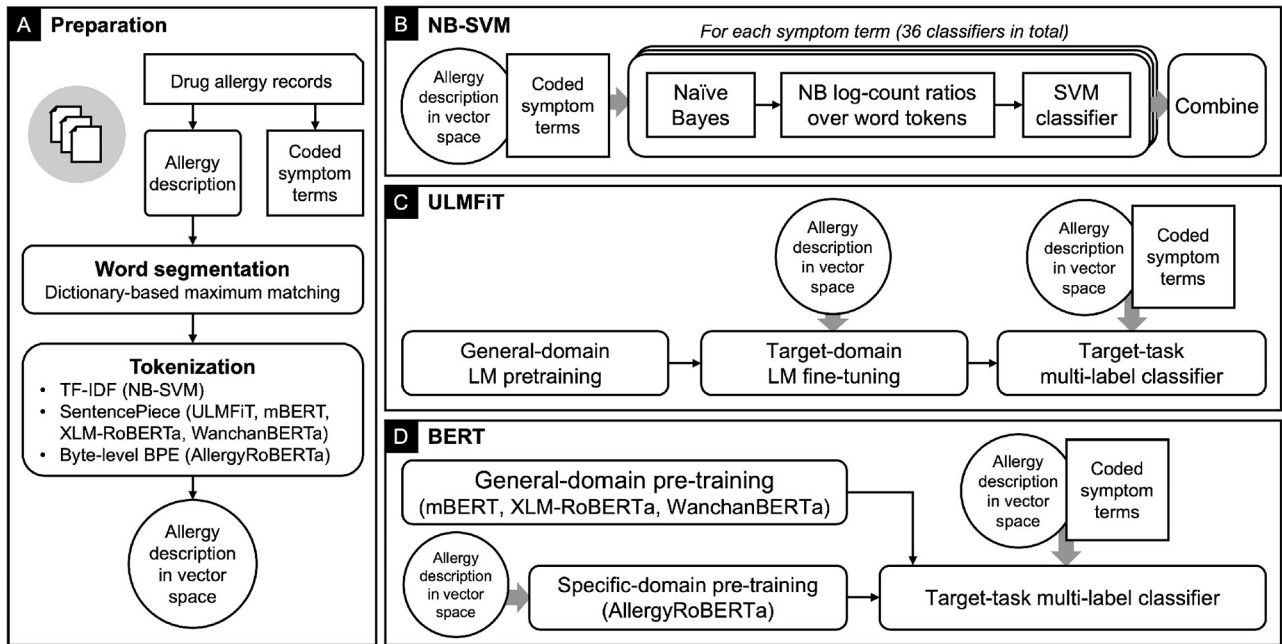

**Fig 3. Diagrams outline the steps involved in our methods.** (A) Data preparation included word segmentation and tokenization, with each algorithm using different method. (B) NB-SVM involved training multiple pipelines for Naive Bayes feature extraction and SVM classification. (C) ULMFiT involved fine-tuning the pre-trained LM with our target-domain allergy corpus and fine-tuning a classifier for our multi-label classification task. (D) BERT involves fine-tuning a classifier for our multi-label classification task. This study evaluated three pre-trained general-domain BERT models and one target-domain BERT model pre-trained on our allergy corpus.

using the vocabulary learned from the training documents. The matrix was then multiplied by the ratios of the NB log counts before being passed to each individual SVM classifier, which in turn produced a prediction for the corresponding class. NB-SVM, however, has the major disadvantage of ignoring unknown words that were not in the training vocabulary. We used the validation set to find the best C parameter for each SVM model. We implemented our models with the scikit-learn module version 0.24.2 on Python version 3.8.10.

**Universal Language Model Fine-tuning (ULMFiT).** Language modelling is a technique in which a sequence of words is analyzed to determine the probability of each word occurring in the sequence. An LM learns the features and characteristics of basic language and uses these learned features to understand new phrases and accurately predict new sequences. LM is used as a base model in various NLP tasks such as text classification, summarization and generation.

ULMFiT is a technique for fine-tuning a LM for specific downstream tasks. The underlying network architecture of ULMFiT is the weight-dropped LSTM architecture (AWD-LSTM) which employs DropConnect to randomly drop a selected subset of weights for regularization and an averaged stochastic gradient descent method (ASGD) for optimization. Our ULMFiT implementation consists of two stages:

First, LM was pre-trained on our corpus in order to learn the properties of the languages. We followed the official implementation of ULMFiT [23]. We used the three-layer AWD-LSTM network proposed by Howard and Ruder [10] with an embedding size of 400 and 1,152 hidden nodes per layer. We employed SentencePiece tokenizer [24] to segment words into subword units for all allergy records in the training set, resulting 7,856 words in our vocabulary. The pre-trained AWD-LSTM network was then augmented for next word prediction by ending the network with a linear layers with a number of outputs equal to the number of words in the vocabulary. The fine-tuning's scheduling policy was set to slanted triangular learning rates (STLR) [10] that first increase linearly and later decay linearly. The pre-trained network was fine-tuned using on a categorical cross entropy-loss with an initial learning rate of 0.001 for 40 epochs.

Second, LM was fine-tuned to our target task. The pre-trained LM from the first stage was augmented with two linear layers with 100 hidden nodes and 36 output nodes, respectively. The model was then fine-tuned to our multi-label classification task with a weighted focal loss [25] with an initial learning rate of 0.001 for 60 epochs. Gradual unfreezing was performed to unfreeze the model starting from the last layer to prevent over-fitting and under-fitting the model.

The focal loss [25] was used to address our class imbalance problem by applying a dynamic weighting factor to the categorical cross-entropy loss to modulate learning by downweighting the contribution of easy-to-classify samples to the total loss and upweighting the contribution of hard-to-classify samples to the total loss. We used the validation set to optimize learning rates and indicate the point at which the training converges. We implemented our ULMFiT model with the fastai library version 2.4.1 [23] on Python version 3.8.10.

**Bidirectional Encoder Representations from Transformers (BERT).** A transformer is a general-purpose architecture for modelling sequential information with the encoder-decoder mechanism. The encoder produces contextual encodings that contain information about the relevance of each part of the input data to the others, while the decoder conversely incorporates contextual information to produce an output sequence. Unlike traditional RNNs that process the inputs in order, transformers employ the attention mechanism that assigns different weights to each part of the input data depending on its importance, so that the model does not have to process the inputs in order. This allows information from one token to propagate

arbitrarily to distant tokens without the vanishing gradient problem. This led to the development of pre-trained approaches from very large natural language datasets.

Bidirectional Encoder Representations from Transformers (BERT) is a transformer-based model for pre-training language representations. BERT has a stack of encoders which are responsible for finding complex relationships between input representations and encoding them in the output. BERT has generally been pre-trained for masked language modelling in which the model was learned to predict a masked token from context, so that BERT can learn contextual embeddings for words. Similar to ULMFiT, BERT pre-training is computationally expensive, but the pre-trained BERT can be fine-tuned for specific tasks on smaller datasets with fewer computational resources. Examples of pre-trained BERT-based models are mBERT (Multilingual BERT) [26] which was pre-trained on 104 languages, XLM-RoBERTa [12] which was pre-trained on a large multi-lingual corpus of 2.5 terabytes, and WangchanBERTa [27] which was pre-trained on assorted Thai texts of 78 gigabytes.

In our implementation, we evaluated three different pre-trained models, i.e. mBERT [26], XLM-RoBERTa [12], and WangchanBERTa [27]), and one RoBERTa-based model [28] that were pre-trained from scratch using our allergy documentation in the training set, called AllergyRoBERTa. We used SentencePiece tokenizer [24] for mBERT [26], XLM-RoBERTa [12], and WangchanBERTa [27], as in their original studies. We used the byte-level byte-pair encoding (BPE) tokenizer [28] for AllergyRoBERTa. We augmented the pre-trained models for our multi-label classification with two linear layers with 768 hidden nodes and 36 output nodes respectively. Similar to ULMFiT, the focal loss [25] was used to tackle class imbalance during training. It dynamically modified the categorical cross-entropy loss to focus more on learning hard negative samples. We fine-tuned our networks using a weighted focal loss with an initial learning rate of $10^{-5}$ for 30 epochs. We used the validation set to determine the best model. We employed the HuggingFace library [29] on Python version 3.8.10.

## Evaluation

In multi-label problems, each instance can belong to several labels; as such, the prediction can be fully correct, partially correct, and fully incorrect [30]. Evaluation metrics that are defined for single-label classification (such as accuracy, precision, and recall)might not be able to capture such notion. For assessing the performance of our models, we used standard metrics for multi-label classification to compare the predicted labels with the ground truth labels: exact match ratio, accuracy, Hamming loss, precision, recall, F1-score, and mean average precision [30]. Exact match ratio is the most strict metric that measures the percentage of samples in which all their labels were correctly classified, while accuracy accounts for partial correctness by calculating the intersection over the union of predicted and ground truth labels. Hamming loss measures the fraction of incorrectly predicted labels out of the total number of labels. Please refer to S2 Appendix for details of each evaluation metric.

After evaluating the performance of each model, we wanted to prospectively test the best-performing algorithm in a similar environment in which we planned to use it. We developed a symptom term suggestion system that retrieved a free-text allergy description from the EHR database and coded it into relevant symptom terms. Pharmacists were presented with the free-text description and the symptom terms suggested by the algorithm. They then coded the symptom terms using the information presented. We measured the agreement between the symptom terms suggested by the model and the symptom terms coded by the pharmacists using Krippendorff's alpha ($\alpha$).

All experiments were performed on a workstation equipped with an NVIDIA Quadro RTX 6000 with 24 gigabtyes of memory.

**Table 3. Performance of each natural language classifier for extracting symptoms from free-text records.**

| Algorithm | Exact Match Ratio | Accuracy | Hamming Loss | Precision | Recall | F₁-score | Mean Average Precision |
|---|---|---|---|---|---|---|---|
| NB-SVM | 89.09 | 95.63 | 0.37 | 97.15 | 97.47 | 96.95 | **97.25** |
| ULMFiT | 91.62 | 96.95 | 0.32 | 97.90 | 98.45 | 97.93 | 94.30 |
| mBERT | 88.81 | 94.38 | 0.55 | 96.22 | 96.22 | 95.77 | 90.36 |
| XLM-RoBERTa | 94.20 | 97.46 | 0.31 | 98.86 | 98.08 | 98.21 | 92.25 |
| WangchanBERTa | 92.88 | 96.62 | 0.40 | 98.30 | 97.41 | 97.53 | 90.14 |
| AllergyRoBERTa | 94.23 | 97.25 | 0.33 | 98.73 | 97.79 | 97.96 | 90.95 |
| Ensemble model | **95.33** | **98.37** | **0.17** | **99.31** | **98.72** | **98.88** | 97.07 |

## Results

This section describes the performance of our models evaluated on the test set of 7,988 samples. Table 3 details the performance of each classifier in extracting symptoms from free text allergy description for each performance metric. We employed three NLP techniques: NB-SVM, ULMFiT, and BERT. Four different BERT-based models were evaluated: mBERT, XLM-RoBERTa, WanchanBERTa, and AllergyRoBERTa. We also evaluated the ensemble model which aggregates the prediction of each model through majority voting and results in the final prediction for measuring against the ground truth labels.

XLM-RoBERTa and AllergyRoBERTa achieved similar results with the highest and second highest scores, respectively, in exact match ratio, accuracy, precision, recall, and F₁ score among the BERT-based models. ULMFiT had higher exact match ratio and accuracy than mBERT but lower in those scores than the other BERT-based models. However, ULMFiT had better recall, F₁ score, and mean average precision than all BERT-based models. NB-SVM has the lowest exact match ratio with moderate scores across different classification metrics but with the highest mean average precision. The ensemble model scored the highest in all performance metrics except for mean average precision.

Table 4 shows the average precision scores of NB-SVM, ULMFiT, XLM-RoBERTa (the highest scoring BERT-based model), and the ensemble model for each symptom term on the test set. Average precision is a single-valued metric based on the ranked list of coded symptom terms returned by the algorithm. It is the average of the precision calculated at each threshold in the ranking list. Generally, NB-SVM performed better for less frequently coded symptom terms, while XLM-RoBERTa performed better for more frequently coded symptom terms. NB-SVM had the highest mean average precision of 97.25% followed by the ensemble model with 97.07%. Table 5 shows the confusion matrix for each symptom term. Misclassification mostly occurs in symptoms related to the integumentary system.

After training, we tested our ensemble model by having the model suggest symptom terms for pharmacists based on a given allergy description. Pharmacists can opt to choose symptom terms suggested by our algorithm and select other relevant symptom terms. Our algorithm can suggest 36 symptom terms, while pharmacists can select from a choice of 45 symptom terms. The Krippendorff's alpha coefficient for comparing symptom terms coded by our ensemble model to pharmacists was 0.7081 on 420 drug allergy records.

## Discussion

This section discusses the relevant findings in light of the study's results, compares them with other related studies, offers suggestions for implementation, and explains limitations as well as future study directions.

**Table 4. Average precision for each symptom term on the test set.**

| Symptom term | NB SVM | ULM FiT | XLM RBT | ENSB |
|---|---|---|---|---|
| **Anaphylaxis A** | | | | |
| Anaphylactic reaction (88) | **100.0** | 99.56 | 98.91 | **100.0** |
| **Anaphylaxis B** | | | | |
| Anaphylactoid reaction (4) | 95.00 | 91.67 | 75.04 | **100.0** |
| Anaphylaxis (100) | **100.0** | 99.26 | 96.94 | **100.0** |
| Shock (55) | **100.0** | 98.22 | 98.89 | **100.0** |
| **Cardiovascular system** | | | | |
| Edema (690) | 98.30 | 98.65 | 98.74 | **99.67** |
| Hypertension (2) | **83.33** | **83.33** | 25.03 | 70.00 |
| Hypotension (15) | **100.0** | 94.24 | 96.78 | 97.68 |
| **Nervous system** | | | | |
| Insomnia (7) | **100.0** | 84.22 | **100.0** | **100.0** |
| **Integumentary system** | | | | |
| Angioedema (2,266) | 99.30 | 99.87 | 99.50 | **99.96** |
| Eczema (37) | **100.0** | 97.10 | 95.51 | 98.21 |
| Erythema multiforme (68) | 72.83 | 79.76 | 78.48 | **82.06** |
| Erythematous rash (1,000) | 98.39 | 98.96 | 97.79 | **99.27** |
| Maculopapular rash (978) | 95.21 | 97.90 | 97.39 | **98.90** |
| Papular rash (699) | 99.31 | 99.91 | 98.20 | **99.99** |
| Pruritus (2,102) | 99.84 | 99.73 | 99.19 | **99.97** |
| Purpura (10) | 87.64 | 94.55 | 95.88 | **100.0** |
| Rash (2,991) | 98.78 | 99.36 | 99.46 | **99.87** |
| SJS/TEN (63) | 99.35 | 99.21 | 85.68 | **100.0** |
| Urticaria (613) | **100.0** | 99.92 | 98.11 | **100.0** |
| **Endocrine system** | | | | |
| Acidosis (2) | **100.0** | **100.0** | **100.0** | **100.0** |
| **Digestive system** | | | | |
| Abdominal pain (57) | 98.85 | 97.99 | 84.60 | **100.0** |
| Diarrhea (39) | **100.0** | **100.0** | 86.62 | 99.56 |
| Nausea (736) | 99.89 | **100.0** | 98.24 | **100.0** |
| Vomiting (437) | **100.0** | **100.0** | 97.84 | **100.0** |
| **Hematopoietic system** | | | | |
| Agranulocytosis (4) | **100.0** | **100.0** | **100.0** | **100.0** |
| Anemia (3) | **100.0** | **100.0** | **100.0** | **100.0** |
| Hemolysis (1) | **100.0** | **100.0** | **100.0** | **100.0** |
| Neutropenia (4) | 95.00 | 66.07 | **100.0** | 91.67 |
| Pancytopenia (2) | **100.0** | **100.0** | **100.0** | **100.0** |
| Thrombocytopenia (2) | **83.33** | 66.67 | 53.57 | 58.33 |
| **Hepatic portal system** | | | | |
| Hepatitis (32) | 99.82 | 97.98 | 99.82 | **100.0** |
| Jaundice (8) | **100.0** | **100.0** | 90.00 | **100.0** |
| **Respiratory system** | | | | |
| Cough (226) | 98.77 | 99.14 | 98.55 | **99.94** |
| Dyspnea (1,063) | 99.88 | 99.94 | 97.79 | **99.98** |
| Respiratory depression (249) | 98.17 | 98.52 | 91.73 | **99.63** |
| **Miscellaneous** | | | | |
| Fever (173) | **100.0** | 99.96 | 86.88 | 99.97 |

(*Continued*)

**Table 4.** (Continued)

| Symptom term | NB SVM | ULM FiT | XLM RBT | ENSB |
|---|---|---|---|---|
| Mean average precision | **97.25** | 94.30 | 92.25 | 97.07 |

* The number in parentheses after each symptom term represents the total number of coded symptom terms in the test set.

## Comparison of different techniques

There is no single metric that provides a comprehensive comparison of different models for multi-label text classification [30]. Overall, BERT-based models (XLM-RoBERTa and AllergyRoBERTa) performed better than ULMFiT. This could be due to the architectural differences between the two models, with BERT having a larger number of model parameters, bidirectional processing, and the attention mechanism [11]. ULMFiT performed better than NB-SVM. This could be due to the language understanding capability embedded in the neural network. The overall model performance of the ensemble model outperformed the performance of each individual technique.

Although DL approaches with recent breakthroughs in text comprehension, such as ULMFiT and BERT, have been suggested for text classification, we observed that the DL approaches outperformed NB-SVM. The highest performance improvement was observed for exact match ratio, which is the proportion of coded symptom terms that exactly matched the ground truth in all terms. XLM-RoBERTa and AllergyRoBERTa had similar highest exact match ratios, followed by ULMFiT and NB-SVM, which were 2.6% and 5.2% lower, respectively. Similar findings applied to accuracy, precision, and $F_1$-score, but with less than 1% or less improvement. We hypothesize that this was because our drug allergy records typically contain short descriptions of about 20–40 words per record, and NB-SVM had been shown to perform reasonably well in text classification of such short documents [9]. In comparison to traditional rule-based and statistical approaches, NLP techniques can obviate the complex and time-consuming processes of creating the dictionary and developing rule-based strategies for text matching.

In term of average precision (see Table 4), although previous work has shown that ULMFiT and BERT-based models outperformed NB-SVM for small datasets [31], we observed otherwise with NB-SVM having higher average precision on several symptom terms with fewer than a hundred items. Compared to NB-SVM, ULMFiT and XLM-RoBERTa had slightly higher average precision on more frequently coded symptom terms, such as angioedema and rash. XLM-RoBERTa performed poorly on symptom terms with few items, such as anaphylectoid reaction, abdominal pain, diarrhea, thrombocytopenia, respiratory depression, and fever. This resulted in XLM-RoBERTa having lower unweighted mean average precision than the other techniques (see Table 4).

We suspect that high average precision values for some symptom terms with few items may be due to an exact match of the symptom term in the free text allergy description. In the integumentary system group (see Table 4), we observed that incomplete allergy description by healthcare professionals may lead to difficulties in coding symptom terms by pharmacists when determining whether rash symptoms fall into which categories (erythematous rash, maculopapular rash, papular rash, purpura, or urticaria). This resulted in slightly lower average precision values for symptom terms in the integumentary system group for all techniques.

One of the aims of this study is to assess the understanding of NLP for multilingual context in the medical domain. Surprisingly, the bilingual context, where both languages were mixed in one unstructured document, was not a problem at all. All NLP techniques can be applied without the need for further processing in our task. One problem we have observed with the

**Table 5. Confusion matrix for each symptom term on the test set.**

| Symptom term | TP[1] | FP[2] | FN[3] | TN[4] |
|---|---|---|---|---|
| **Anaphylaxis A** | | | | |
| Anaphylactic reaction (88) | 88 | 0 | 0 | 7,904 |
| **Anaphylaxis B** | | | | |
| Anaphylactoid reaction (4) | 2 | 0 | 2 | 7,988 |
| Anaphylaxis (100) | 98 | 0 | 2 | 7,892 |
| Shock (55) | 53 | 0 | 2 | 7,937 |
| **Cardiovascular system** | | | | |
| Edema (690) | 657 | 5 | 33 | 7,297 |
| Hypertension (2) | 0 | 0 | 2 | 7,990 |
| Hypotension (15) | 13 | 0 | 2 | 7,977 |
| **Nervous system** | | | | |
| Insomnia (7) | 5 | 0 | 2 | 7985 |
| **Integumentary system** | | | | |
| Angioedema (2,266) | 2,224 | 5 | 42 | 5,721 |
| Eczema (37) | 33 | 0 | 4 | 7,955 |
| Erythema multiforme (68) | 60 | 16 | 8 | 7,908 |
| Erythematous rash (1,000) | 980 | 27 | 20 | 6,965 |
| Maculopapular rash (978) | 932 | 43 | 46 | 6,971 |
| Papular rash (699) | 696 | 4 | 3 | 7,289 |
| Pruritus (2,102) | 2,087 | 12 | 15 | 5,878 |
| Purpura (10) | 9 | 0 | 1 | 7,982 |
| Rash (2,991) | 2,882 | 8 | 109 | 4,993 |
| SJS/TEN (63) | 57 | 0 | 6 | 7,929 |
| Urticaria (613) | 606 | 0 | 7 | 7,379 |
| **Endocrine system** | | | | |
| Acidosis (2) | 2 | 0 | 0 | 7,990 |
| **Digestive system** | | | | |
| Abdominal pain (57) | 49 | 0 | 8 | 7,935 |
| Diarrhea (39) | 33 | 0 | 6 | 7,953 |
| Nausea (736) | 731 | 0 | 5 | 7,256 |
| Vomiting (437) | 433 | 0 | 4 | 7,555 |
| **Hematopoietic system** | | | | |
| Agranulocytosis (4) | 4 | 0 | 0 | 7,988 |
| Anemia (3) | 3 | 0 | 0 | 7,989 |
| Hemolysis (1) | 1 | 0 | 0 | 7,991 |
| Neutropenia (4) | 3 | 0 | 1 | 7,988 |
| Pancytopenia (2) | 2 | 0 | 0 | 7,988 |
| Thrombocytopenia (2) | 1 | 0 | 1 | 7,990 |
| **Hepatic portal system** | | | | |
| Hepatitis (32) | 31 | 0 | 1 | 7,960 |
| Jaundice (8) | 7 | 0 | 1 | 7,984 |
| **Respiratory system** | | | | |
| Cough (226) | 218 | 0 | 8 | 7,766 |
| Dyspnea (1,063) | 1052 | 1 | 11 | 6,928 |
| Respiratory depression (249) | 244 | 2 | 5 | 7,741 |
| **Miscellaneous** | | | | |

(*Continued*)

**Table 5.** (Continued)

| Symptom term | TP[1] | FP[2] | FN[3] | TN[4] |
|---|---|---|---|---|
| Fever (173) | 164 | 1 | 9 | 7,818 |

[1] TP: True Positive;

[2] FP: False Positive;

[3] TP: False Negative;

[4] TN: True Negative

techniques based on LM is that the original vocabulary contained in the pre-trained models can be very small. This could hinder language comprehension in a medical context. One could develop LM that is specifically pre-trained on a large corpus of clinical notes in an institutional EHR. This could also facilitate the development of other internal NLP algorithms that are aware of the local context.

## Comparison of different BERT models

BERT achieves self-supervised learning by masking and distorting a few words in each sentence and learning the model by guessing those words. A pre-trained BERT model can be used without labeled data for language representation, with the learned vectors constituting BERT's vocabulary and the capability of the model to fill in the blanks with context-sensitive vectors neighboring to masked words.

The most common application of a pre-trained BERT model is to fine-tune for a downstream supervised task. An implicit assumption is made that the pre-trained model was trained properly and performs appropriately. Only the performance of a fine-tuned model is evaluated; no assumption is further made about the quality of the pre-trained model. Sometimes, fine-tuning may be preceded by continual pre-training to improve the performance of the model. Model training, whether it is pre-training, continuous pre-training, or fine-tuning, affects both the model weights and sometimes the vocabulary vectors.

However, continual pre-training, which involves the training the original BERT models further on a domain-specific corpus while retaining the original vocabulary, may not result in significant performance improvement due to the uniqueness of medical domain languages. Such a model's performance may still lag behind a model that is pre-trained from scratch on a domain-specific corpus with a domain-specific vocabulary.

In the medical domain, there are a variety of unique terms or phrases, such as drug names, symptoms, and diseases. Most general-domain BERT models have been pre-trained with a vocabulary heavily slanted toward people, places, and organizations. There are other sentence fragments or patterns that are exclusive to the medical domain In particular, in our problem, clinical notes were presented intermixed in both languages.

Although many studies have shown that biomedical and clinical LMs (such as BioBERT [13], ClinicalBERT [14], and BioALBERT [15]) performed better on downstream clinical tasks than general-domain LMs, all available domain-specific LMs have been pre-trained with a monolingual English corpus only. It is not possible to finetune these LMs with our institutional non-English dataset and compare their performances.

Despite the fact that our domain-specific corpus is much smaller than the general-domain corpus, training the model from scratch on our domain-specific allergy corpus with a custom vocabulary generated from the corpus itself has been shown to be critical for maximizing model performance. AllergyRoBERTa, which was trained on just 1.2 million tokens, achieved

nearly on par performance for our downstream multi-label classification task in comparison with XLM-RoBERTa, which was pre-trained 2.5 terabytes of multi-lingual text comprising billions of tokens. AllergyRoBERTa also surpassed the other general-domain BERT models, i.e. mBERT and WanchanBERTa, which was pre-trained on smaller number of tokens. This could be primarily due to the particular linguistic features of the medical domain, which are typically underrepresented in the original general-domain pre-trained models. We hypothesize that XLM-RoBERTa is a good choice to perform downstream fine-tuning for a non-English target task when a large domain-specific corpus is not available for pre-training or continuous pre-training of the LM. In the English clinical context, it is recommended to explore biomedical and clinical LMs, such as BioBERT [13], ClinicalBERT [14], and BioALBERT [15].

## Comparison to other studies

The extraction of clinical symptoms from unstructured clinical texts has been investigated in many studies [5, 16–20]. The standardized symptomatic terms resulted from these works have been used for epidemiological research, clinical systems improvement, and healthcare interoperability. It is, however, difficult to make direct comparison to other studies because each study used its own institutional dataset and had its own unique purpose.

Compared with previous studies that used rule-based techniques [5, 16, 17, 19], our techniques were developed without the use of data dictionaries or clinical concept modeling. In the rule-based studies, Wagholikar *et al.* [16] created a script for mapping various synonyms and acronyms to SNOMED CT concepts. The approach was accurate for specific symptoms such as chest pain but not for more complex groups such as stomach pain and trauma. Epstein *et al.* [5] implemented an allergy-matching algorithm with Transact-SQL that included seven lookup tables. The algorithm achieved an $F_1$ score of 0.98. The authors pointed out that periodically checking the remaining entries not found by the algorithm and periodically adding terms to the lookup tables could help improve the algorithm. Goss *et al.* [17] compiled several lists of standard terminologies with a set of rules defined for encoding allergy concepts. The authors noted a need for the algorithm to consider a contextual understanding between allergens and symptoms for best results. Recently, the rule-based pipeline that developed by Lenivtceva *et al.* [19] employed various techniques to handle misspellings and abbreviations, implemented over 20 regular expression rules, and constructed over four dictionaries of over 2,675 terms to describe allergy categories in Russian. Thanks to the complicated and specialized approach, the authors reported very good performance with $F_1$-score of 0.90–0.96 for allergy categorization. Implementing the rule-based approach could be time-consuming and tedious, but it is not necessary to have a well-curated large dataset.

The ML/DL approach has been reported to perform well, comparable to or better than rule-based methods, but depends heavily on the quality of the data. In practice, the ML/DL approach requires a well-curated dataset. Classical ML algorithms, such as SVM, have been employed in Jackson *et al.* [18] that involved human annotators to label clinical documents to in order to create training corpora. The approach can extract symptoms from clinical text with a median $F_1$ score of 0.88 across 46 symptoms. In Leiter *et al.* [20], a DL algorithm was developed to identify symptoms from clinical notes of patients with congestive heart failure. The algorithm had an $F_1$ score of 0.72 which might be due to a small sample size of only 154 notes.

Our study benefits from the availability of data that have already been labelled by experts over time. Without the need to define explicit rules, the ML/DL approach can reduce the time and effort required to develop clinical NLP algorithms. In addition, with a large training corpus, the ML/DL approach can learn some degree of noise in the text data, e.g. abbreviations, misspellings, uncertainties, and variations in clinical notes. However, the ML/DL approach

may perform poorly on certain underrepresented symptom groups because the algorithm learns the categorization of the text from the data rather than from explicit rules.

## Toward implementation

Our work can provide context-aware suggestion of relevant symptom terms based on unstructured drug allergy description. It can be used to make suggestions during clinical documentation leading to more complete and systematic documents. We anticipate that such implementation will allow healthcare professionals to provide relevant symptom terms at hand when creating a new drug allergy record, resulting in more complete reporting and thus reducing delays and effort in review by pharmacists. It could also lead to faster alerts when prescribing medications through the medication verification system, in the event that the patient is prescribed a medication to which they are allergic.

In our test of agreement with pharmacists, we had a Krippendorff's alpha coefficient of 0.7081. Since Krippendorff's alpha compares the observed disagreement with the expected disagreement, the alpha tends to penalise variables with highly skewed distributions. In our case, the coefficient tends to be low because we have multiple class labels with highly skewed distributions.

The implementation of DL-based approaches may require an access to algorithms on high-performance servers, as such users may experience processing delays. NB-SVM is not resource intensive and can be easily implemented offline, but with lower performance, in the original drug allergy reporting module within the EHR.

## Limitations

This study was subject to several limitations. First, our experiments were conducted with a large but highly biased institutional dataset. Many symptom terms have been coded less than 100 times in the past two decades. We used class weighting and focal loss to deal with class imbalance. We did not perform data balancing, such as oversampling or undersampling, as this is difficult to perform on multilingual context and further assumptions must be made for such balancing. Second, we did not investigate different tokenization methods; instead, the standard tokenization suggested in the original literature was used for each technique. With a more sophisticated ways of learning text representation, such as BPE, which can handle words outside the vocabulary and inflected words, further performance gains could be observed. Third, our study was based on the local context of our institution where we developed our in-house EHR. Different institutions may have different methods for documenting drug allergy in their system, some with more or fewer data fields. Our algorithms may not generalize well to different contexts.

## Conclusion

The systematic representation of drug allergy is critical for patient safety, clinical research, and quality improvement. While allergy information is routinely collected by healthcare professionals at every stage of care, it is frequently inadequately documented in the EHR. Despite the fact that many EHR systems support structured reporting, free-text drug allergy reporting is still common.

This study aims to develop algorithms for mapping free-text drug allergy records to standard symptom terms. We used a large dataset of 79,912 verified drug allergy records collected from our institutional EHR. We evaluated three different NLP methods: NB-SVM, ULMFiT, and BERT. All techniques performed adequately in our bilingual context without the need for advanced pre-processing. Overall, BERT performed better than ULMFiT, and ULMFiT

performed better than NB-SVM. For BERT, multilingual models pre-trained with more data performed better than those pre-trained with less data. The performance of XLM-RoBERTa pre-trained with very large general-domain data was slightly exceeded with AllergyRoBERTa pre-trained from scratch with our drug allergy reports. With the ensemble model, we achieved an exact match ratio of 95.33%, a $F_1$ score of 98.88%, and a mean average precision of 97.07% for the 36 most frequently coded symptom terms. We developed a simulated environment to examine the integration to EHR and investigate the agreements of symptom terms suggested by our algorithm against those coded by pharmacists on 420 drug allergy records. We obtained a Krippendorff's alpha agreement coefficient of 0.7081.

The ability of NLP to accurately capture both semantic and syntactic structures in clinical documents is becoming increasingly important for the efficient extraction of structured clinical information. The structured information stored in EHR can be used for other applications, such as clinical decision support systems and prescription alerts. It also enables healthcare interoperability so that clinical information can be easily shared between organizations using standard terminology with different EHRs.

## Supporting information

**S1 Appendix. Naive Bayes—Support Vector Machine (NB-SVM).**
(PDF)

**S2 Appendix. Evaluation metrics [32].**
(PDF)

## Acknowledgments

This study was approved by the Human Research Ethics Committee (HREC), Faculty of Medicine, Prince of Songkla University (REC. 64-100-25-2) with a waiver for written informed consent due to its retrospective nature. We acknowledge the support of computational resources from the Division of Digital Innovation and Data Analytics (DIDA), which was in turn supported by the Kasikornthai Foundation and the SCG Foundation. The authors have no conflicts of interest to declare.

## Author Contributions

**Conceptualization:** Sitthichok Chaichulee, Chanon Kongkamol, Pasuree Sangsupawanich.

**Data curation:** Sitthichok Chaichulee, Chissanupong Promchai, Tanyamai Kaewkomon, Chanon Kongkamol, Thammasin Ingviya.

**Formal analysis:** Sitthichok Chaichulee, Chissanupong Promchai, Tanyamai Kaewkomon, Thammasin Ingviya.

**Investigation:** Sitthichok Chaichulee, Chissanupong Promchai, Tanyamai Kaewkomon, Pasuree Sangsupawanich.

**Methodology:** Sitthichok Chaichulee.

**Project administration:** Pasuree Sangsupawanich.

**Resources:** Chanon Kongkamol.

**Software:** Sitthichok Chaichulee.

**Supervision:** Pasuree Sangsupawanich.

**Validation:** Chissanupong Promchai, Tanyamai Kaewkomon, Chanon Kongkamol, Thammasin Ingviya.

**Visualization:** Sitthichok Chaichulee.

**Writing – original draft:** Sitthichok Chaichulee, Chissanupong Promchai, Tanyamai Kaewkomon, Pasuree Sangsupawanich.

**Writing – review & editing:** Sitthichok Chaichulee.

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
