## [Decision Letter · Decision Letter 0]

20 Jan 2022

PONE-D-21-35704Multi-label classification of symptom terms from free-text bilingual drug allergy records using natural language processingPLOS ONE

Dear Dr. Chaichulee,

Thank you for submitting your manuscript to PLOS ONE. After careful consideration, we feel that it has merit but does not fully meet PLOS ONE’s publication criteria as it currently stands. Therefore, we invite you to submit a revised version of the manuscript that addresses the points raised during the review process.

We look forward to receiving your revised manuscript.

Kind regards,

Junaid Rashid, Ph.D

Academic Editor

PLOS ONE

Journal Requirements:

3. Please ensure that you refer to Figure 1 in your text as, if accepted, production will need this reference to link the reader to the figure.

4. We note you have included a table to which you do not refer in the text of your manuscript. Please ensure that you refer to Table 3 in your text; if accepted, production will need this reference to link the reader to the Table.

Reviewers' comments:

Reviewer's Responses to Questions

**Comments to the Author**

1. Is the manuscript technically sound, and do the data support the conclusions?

Reviewer #1: Yes

Reviewer #2: Partly

2. Has the statistical analysis been performed appropriately and rigorously? 

Reviewer #1: Yes

Reviewer #2: Yes

3. Have the authors made all data underlying the findings in their manuscript fully available?

Reviewer #1: No

Reviewer #2: No

4. Is the manuscript presented in an intelligible fashion and written in standard English?

Reviewer #1: Yes

Reviewer #2: Yes

5. Review Comments to the Author

Reviewer #1: In this paper, authors evaluated and compared three types of NLP methods and their variations, Naive Bayes - Support Vector Machine (NB-SVM), Universal Language Model Fine-tuning (ULMFiT), and Bidirectional Encoder Representations from Transformers (BERT), for classification of symptom terms in English and Thai. The paper is well written, the proposed methods are briefly reviewed, and the results are clearly presented. As expected, the comparison results showed that the BERT-based methods perform better in general and the ensembled method performs the best among all different methods for different evaluation criteria. The main concern is lacking of novelty:

• Need more justifications for methodological novelty if any. In particular, the two different languages (Thai and English) need to be dealt with. What is new to use the proposed NLP methods to deal with two languages simultaneously?

• Need to discuss more details on the novel clinical findings if any.

Reviewer #2: Thank you so much for sharing your manuscript. The authors of "Multi-label classification of symptom terms from free-text bilingual drug allergy records using natural language processing" evaluate the ability of several natural language models to predict symptoms from unstructured texts from electronic health records. I commend the authors on their creation of an allergy-specific BERT-based model and focus on an important clinical construct. However, there are key portions of the manuscript that are not clear, which make it difficult to understand why certain decisions were made. More details are included below, along with a few additional questions. I hope that these comments can help to strengthen the manuscript.

General Comments:

- I appreciate the inclusion of Figures 1 and 3 to try and explain the experimental pipeline. However, while I can generally understand what was done for the “Data Preparation” and “Algorithm Development,” the “Evaluation” process is not clear. For example, the data is split into three parts for training, validation, and testing, but I am unclear on what kind of model was used for “training” or what hyperparameters were “validated.” Furthermore, you discuss precision, recall, and accuracy, but I am unclear on how gold-standard labels were obtained. Please clarify this information within the text.

- Right now, you compare the natural language models to each other to identify the one with the best performance. While the metrics obtained appear impressive, it would be helpful if you compare your results to a simple text-matching model. If the simple method performs poorly compared to the models presented already, it supports the need for more complex models.

- You note that the data is imbalanced, yet no correction is implemented. As a result, I am skeptical of the high performance reported in Table 3 and concerned that not correcting for this imbalance may limit the generalizability of the results. Please consider running a sensitivity analysis where you over- or under-sample the training data and report those results alongside the non-adjusted data. In addition, please consider including a confusion matrix so that readers can see which symptoms are typically are mistaken for each other (this may also help with some interpretation of what your model is “learning”).

Specific Comments:

Abstract

- The abstract first mentions three different “NLP techniques” (i.e., NB-SVM, ULMFiT, BERT) but then names different models in the following sentence (XLM-RoBERTa, AllergyRoBERTa). While this makes sense after reading the article, it is not does not make sense without that additional information. Please clarify in the abstract that you tested different BERT-like models that include XLM-RoBERTa and AllergyRoBERTa.

Introduction

- Lines 10 – 11: There are several phrases within the Introduction (e.g., “cutaneous manifestations”) that could be considered clinical jargon. Since the audience of this journal is broad, please consider rephrasing some of this language or providing a description that is understandable to non-health audiences.

Methods

- “Data Collection”, Lines 186 – 187: You mention that you obtain data for 18 years, yet only 79,912 records are included. This value feels very small relative to the time window of data collection. It may be helpful to provide additional context to assure the reader that this value is “okay.” Were these records a sample of a broader pool of labels, is the hospital small (and thus does not have that many patients per year), are drug allergy records uncommon, or is there another reason for this?

- “Data Preparation” section: You mention that the free text contains both English and Thai terms, but I am still unclear on how this was accounted for in the model. Were there separate models for each language, or were they combined? Were they evaluated together or separately? Please elaborate.

- “Model development”: While I appreciate the detail in this section, I think that there is actually too much information in the subsections for each model. In an already-complex paper, adding the extreme specifics of how each model works may be lost on readers from general audiences. Please consider moving portions of this section to a Methods supplement so that readers who are interested can read more, but make sure to keep enough in the main paper that it can be understood.

- “Evaluation – Performance metrics”: Consider moving the mathematical descriptions of each of the different metrics to a Methods supplement, as a general audience likely will not understand this and get lost in the details.

Results

- Lines 430 – 432: This information could go in the “Evaluation” section of the methods instead of the Results.

- Lines 441 – 443: I am confused on why Krippendorff’s alpha was used in this study if you are already assuming that the labels provided by the pharmacists are the gold standard.

Discussion

- “Bilingual representation”: Because there were no results presented specifically for a bilingual model (or, if they were, they were done in a way that was not clear), this section seems more speculative than substantive. Please clarify in the methods how the bilingual component of the models were assessed, then describe this outcome in the Results. That way, this portion of the discussion will make more sense.

6. PLOS authors have the option to publish the peer review history of their article (what does this mean?). If published, this will include your full peer review and any attached files.

Reviewer #1: No

Reviewer #2: No

---

## [Author Response · Author response to Decision Letter 0]

28 Feb 2022

Dear Academic Editor and Reviewers,

Thank you for your very thorough review of the manuscript. We greatly appreciate the reviewers' constructive comments, as well as the Academic Editor's decision to allow us to revise and resubmit unclear or overlooked points in our manuscript. We tried to maintain our original motivation and framework for developing the manuscript while correcting and improving the details and unclear statements based on the comments given. We have addressed all comments as described below and in the attached document. Our responses are as follows and are highlighted in yellow in the revised manuscript. We believe we have greatly improved the article and more clearly explained our findings.

Sitthichok

Sitthichok Chaichulee DPhil

Department of Biomedical Sciences and Biomedical Engineering

Faculty of Medicine, Prince of Songkla University 90110, Thailand. 

1. Please ensure that your manuscript meets PLOS ONE's style requirements, including those

for file naming. The PLOS ONE style templates can be found at https://journals.plos.org/

plosone/s/file?id=wjVg/PLOSOne_formatting_sample_main_body.pdf and https://journals

.plos.org/plosone/s/file?id=ba62/PLOSOne_formatting_sample_title_authors_affiliations.pdf

Answer

--------

We have used the PLOS LaTex template as provided in the official website: https://journals .plos.org/plosone/s/latex. We have reformatted the citation of figures and tables according to the guidelines. We have ensured that the order of figure and table placements is after the paragraph in which they are first cited.

Answer

--------

We have included the full ethics statements in the Methods section on Page 6 Lines 173 – 176: “This study was approved by the Human Research Ethics Committee (HREC), Faculty of Medicine, Prince of Songkla University (REC. 64-100-25-2) with a waiver for written informed consent due to its retrospective nature.”

3. Please ensure that you refer to Figure 1 in your text as, if accepted, production will need this reference to link the reader to the figure.

Answer

--------

We apologize that we did not refer Figure 1 in the main content. We have now fixed the issue. All figures and tables that are included in the manuscript are now cited accordingly in the main text.

4. We note you have included a table to which you do not refer in the text of your manuscript. Please ensure that you refer to Table 3 in your text; if accepted, production will need this reference to link the reader to the Table.

Answer

--------

All figures and tables that are included in the manuscript are now cited accordingly in the main text.

Reviewers' comments:

Reviewer's Responses to Questions

Comments to the Author

1. Is the manuscript technically sound, and do the data support the conclusions?

Reviewer #1: Yes

Reviewer #2: Partly

2. Has the statistical analysis been performed appropriately and rigorously? 

Reviewer #1: Yes

Reviewer #2: Yes

3. Have the authors made all data underlying the findings in their manuscript fully available?

Reviewer #1: No

Reviewer #2: No

4. Is the manuscript presented in an intelligible fashion and written in standard English?

Reviewer #1: Yes

Reviewer #2: Yes

 

5. Review Comments to the Author

Reviewer #1:

[R1.1]

In this paper, authors evaluated and compared three types of NLP methods and their variations, Naive Bayes - Support Vector Machine (NB-SVM), Universal Language Model Fine-tuning (ULMFiT), and Bidirectional Encoder Representations from Transformers (BERT), for classification of symptom terms in English and Thai. The paper is well written, the proposed methods are briefly reviewed, and the results are clearly presented. As expected, the comparison results showed that the BERT-based methods perform better in general and the ensembled method performs the best among all different methods for different evaluation criteria. The main concern is lacking of novelty:

• Need more justifications for methodological novelty if any. In particular, the two different languages (Thai and English) need to be dealt with. What is new to use the proposed NLP methods to deal with two languages simultaneously?

• Need to discuss more details on the novel clinical findings if any.

Answer

--------

Thank you for kindly reviewing the article. We appreciate your comments very much. 

In our work, we present a comprehensive pipeline for developing a multi-label classification tool of symptom terms from free-text description that consists of three phases: (1) the collection of drug allergy records from the EHR; (2) the evaluation of different NLP models (NBSVM, ULMFiT, and BERT -based models); and (3) the testing of the best-performing algorithm in a simulated EHR environment in which the algorithm suggests symptom groups from the free-text description, i.e. augmented intelligence. Our results show that our algorithm could help pharmacists in coding symptoms.

Our novelty relies on the investigation on the applicability of NLP in a domain-specific context, i.e., drug allergy, where there are still gaps in the development of such algorithms, especially in an environment where English is not the native language, as we pointed out in the Introduction section. In our drug allergy context, we generally explain clinical symptoms using free-text descriptions that are a mixture of Thai and English words. 

We kindly agree that we have adopted the standard conventional approaches that are really mature for such implementation. We, nevertheless, provided a comprehensive comparison of different algorithms ranging from straightforward NB-SVM to LSTM-based ULMFiT and to newer transformer-based BERT models. We tested two multilingual BERT models: mBERT (pretrained on entire Wikipedia dump of 104 languages) and XLM-ROBERTa (pre-trained on assorted 2.5TB data), one bilingual model: WangchanBERTa (pre-trained on assorted Thai-English 78.5GB data), and our AllergyROBERTa (pre-trained from scratch on our domain-specific allergy corpus). Our results suggest that XLM-RoBERTa had the highest performance, closely followed by AllergyRoBERTa. Our investigations lead to an insight in which XLM-RoBERTa that was pre-trained on a very large multilingual general domain corpus performs well in a clinical context in which pre-trained LM is not available or data for training a new LM is lacking.

To emphasize our contributions, we have added the following paragraphs on Page 5 Lines 153 - 156:

“Our contributions were on the investigation on the applicability of recent NLP approaches in a domain-specific clinical context, i.e. drug allergy, where there are still gaps in the development of such algorithms, especially in a setting where clinical documentation is multilingual.”

NLP can introduce some degree of automation that could both lessen the effort for human coding and speed up the availability of real-time data, leading to real-time alerts. The ability to capture both semantic and syntactic structures in clinical reports, especially in a multilingual context, could lead to more efficient and accurate clinical documentation. We plan to explore other NLP applications in other clinical contexts in our next studies to come.

Reviewer #2:

[R2.1]

Thank you so much for sharing your manuscript. The authors of "Multi-label classification of symptom terms from free-text bilingual drug allergy records using natural language processing" evaluate the ability of several natural language models to predict symptoms from unstructured texts from electronic health records. I commend the authors on their creation of an allergy-specific BERT-based model and focus on an important clinical construct. However, there are key portions of the manuscript that are not clear, which make it difficult to understand why certain decisions were made. More details are included below, along with a few additional questions. I hope that these comments can help to strengthen the manuscript.

Answer

--------

Thank you for your very thorough review of the manuscript. We appreciate your comments very much. We have improved the article in many aspects and explained our contributions more clearly.

[R2.2] 

General Comments:

- I appreciate the inclusion of Figures 1 and 3 to try and explain the experimental pipeline. However, while I can generally understand what was done for the “Data Preparation” and “Algorithm Development,” the “Evaluation” process is not clear. For example, the data is split into three parts for training, validation, and testing, but I am unclear on what kind of model was used for “training” or what hyperparameters were “validated.” Furthermore, you discuss precision, recall, and accuracy, but I am unclear on how gold-standard labels were obtained. Please clarify this information within the text.

Answer

--------

Thank you very much for the comment. At our institution, drug allergy can be reported by physicians, nurses, and pharmacists at any stage of care. The report includes a generic name (single selection, required), trade name (free-text), description (free-text, required), and symptom terms (multiple selection). After reporting, each drug allergy record is subject to be reviewed by pharmacists, who can modify the entry and, if necessary, conduct further investigation. Our study included the drug allergy records that have been flagged as ‘reviewed by pharmacists’. Our gold-standard labels are symptom terms that already verified by pharmacists.

We have revised the content on Page 6 Lines 181 – 183 and Lines 188 – 191.

In Figure 1, we have changed the title from “Data Preparation” to “Data Collection” and “Evaluation” to “Test with Pharmacists”. For the “Data Collection” block, we simply collected the drug allergy records from the EHR database. For the “Algorithm Development” block, we divided the data into three parts for training, validation, and testing, as indicated this in Page 7 Lines 224 - 229. We developed and tuned models on the training and validation sets, and tested them on the test set. For NB-SVM, we used the validation set to find the best C parameter for SVM. For ULMFiT and BERT, we used the validation set to optimize learning rates and indicate the point at which the training converges. The Results section indicates the performance of our models on the test set.

We have revised the content on Page 5 Lines 165 – 170, Page 7 Lines 224 – 229, Page 9 Lines 264 - 265, Page 10 Lines 297 – 298, and Page 11 Lines 335 - 336.

In Figure 3, for simplicity, we have not shown the process by which the dataset was split here. The "Data Preparation" block indicates how a free text allergy description is converted into a sequence of word IDs. Each other block shows the process by which a multi-label classifier is developed.

[R2.3]

- Right now, you compare the natural language models to each other to identify the one with the best performance. While the metrics obtained appear impressive, it would be helpful if you compare your results to a simple text-matching model. If the simple method performs poorly compared to the models presented already, it supports the need for more complex models.

Answer

--------

In order to compare our results with a simple text matching model (e.g. regular expression matching), there is a need to create a dictionary containing words associated with each symptom in both Thai and English. In addition, some strategies need to be implemented so that the model can better categorize different skin rashes as well as symptoms that are similar but have different names because they occur in different organs (e.g., edema and angioedema). Due to a time constraint and the need to establish a panel of experts to discuss the topic and create the dictionary, it is difficult to accomplish in a short period of time.

What we would like to emphasise in our manuscript is that NLP techniques can obviate the need for such complex and time-consuming processes of creating the dictionary and developing strategies for text matching by using the data (free-text allergy description and coded symptom labels) that we already have in our hospital database. The resulting NLP model could be used to improve the functionality of the hospital information system and reduce the time required to code free-text description to symptom terms.

We have improved our manuscript by emphasising the benefit of NLP over a dictionary-based text matching method at the Discussion section on Page 13 - 14 Lines 420 - 423.

[R2.4]

- You note that the data is imbalanced, yet no correction is implemented. As a result, I am skeptical of the high performance reported in Table 3 and concerned that not correcting for this imbalance may limit the generalizability of the results. Please consider running a sensitivity analysis where you over- or under-sample the training data and report those results alongside the non-adjusted data. In addition, please consider including a confusion matrix so that readers can see which symptoms are typically are mistaken for each other (this may also help with some interpretation of what your model is “learning”).

Answer

--------

For the NN-based models, we addressed the data imbalance problem with focal loss, as indicated in Page 10 Lines 294 - 300 for ULMFiT and Page 11 Lines 332 - 334 for BERT. The leverage of class weights in the focal loss during the model fitting process results in the minority classes receiving more attention compared to the majority classes. As a result, during the backpropagation, more loss value is associated with the minority class and the model will give equal attention to all the classes present in the output. For NB-SVM, we addressed the data imbalance problem with the class weighting scheme, as indicated in Page 9 Lines 256 - 259. The parameter C is weighted by the size of the classes, so we penalize SVM more harshly for misclassifying the less frequent class (positive samples) than for misclassifying the more frequent class (negative samples).

For a sensitivity analysis with oversampling and undersampling of the training data, we tried the undersampling of the training data and obtained overfitted models. This might be due to a small number of minority classes. Unlike image analysis that we can perform data augmentation to increase the sample size by randomly flipping, shifting, rotating, and etc. For NLP, we shuffle sentences, add random sentences or words, remove some sentences or words, etc. In Thai, we do not have a dot to separate a sentence, but just use context and grammatical structures to separate one sentence from another. The library for such augmentation in Thai is still immature, both in terms of grammar and syntactic structures, and it is still difficult to get it right, apart from doing it randomly or simply making multiple copies of each entry. We have tested oversampling and also obtained overfitted models.

Although class imbalance may be noted as a main obstacle that prevents classification algorithms to generalize well from the training data to the test data, we presented a comprehensive performance metrics to illustrate the generalization of our models in our manuscript. As suggested, we have included a multi-label confusion matrix so readers can see which symptoms are typically confused with each other on Page 14.

In addition, our test in the simulated environment in which the model is implemented to help pharmacists by suggests relevant terms from free-text descriptions has demonstrated in a good agreement between the coded symptom terms and suggested symptom terms. This suggested the generalizability of the model.

[R2.5]

Specific Comments:

Abstract

- The abstract first mentions three different “NLP techniques” (i.e., NB-SVM, ULMFiT, BERT) but then names different models in the following sentence (XLM-RoBERTa, AllergyRoBERTa). While this makes sense after reading the article, it is not does not make sense without that additional information. Please clarify in the abstract that you tested different BERT-like models that include XLM-RoBERTa and AllergyRoBERTa.

Answer

--------

Thank you very much for the comment. We have revised the abstract to include the following phase: “We tested different general-domain pre-trained BERT models, including mBERT, XLM-RoBERTa, and WanchanBERTa, as well as our domain-specific AllergyRoBERTa, which was pre-trained from scratch.” 

[R2.6]

Introduction

- Lines 10 – 11: There are several phrases within the Introduction (e.g., “cutaneous manifestations”) that could be considered clinical jargon. Since the audience of this journal is broad, please consider rephrasing some of this language or providing a description that is understandable to non-health audiences.

Answer

--------

Thank you very much for your comments. We have revised the clinical part in the manuscript on Page 2 Lines 11 – 17 to make them more understandable to non-medical audiences.

[R2.7]

Methods

- “Data Collection”, Lines 186 – 187: You mention that you obtain data for 18 years, yet only 79,912 records are included. This value feels very small relative to the time window of data collection. It may be helpful to provide additional context to assure the reader that this value is “okay.” Were these records a sample of a broader pool of labels, is the hospital small (and thus does not have that many patients per year), are drug allergy records uncommon, or is there another reason for this?

Answer

--------

Thank you very much for the comment. Although drug allergies are common and all patients are asked before prescribing, many patients do not know their allergy status or have never experienced a drug allergy. The phenomenon of underreporting in pharmacovigilance is well known. 79,912 records are the total number of adverse drug reaction entries that our institution have recorded over the past 18 years. Note that our institution has not implemented hospital information exchange, so allergy data are only collected from our institution.

Underreporting of adverse drug reactions is common and poses a significant challenge to pharmacovigilance [1, 2]. This is because most countries primarily follow the spontaneous or voluntary method of reporting drug allergies.

We have revised the content on Page 6 Lines 176 – 177 and Lines 197 – 299.

References

1 Pushkin R, Frassetto L, Tsourounis C, Segal ES, Kim S. Improving the reporting of adverse drug reactions in the hospital setting. Postgrad Med. 2010;122:154–64.

2. Hazell L, Shakir SA. Under-reporting of adverse drug reactions: A systematic review. Drug Saf. 2006;29:385–96.

[R2.8]

- “Data Preparation” section: You mention that the free text contains both English and Thai terms, but I am still unclear on how this was accounted for in the model. Were there separate models for each language, or were they combined? Were they evaluated together or separately? Please elaborate.

Answer

--------

In our context of drug allergies, we generally explain clinical symptoms with free-text descriptions that are a mixture of Thai and English words. Therefore, our dictionary contains both Thai and English words. For each algorithm, we trained one model that received a free-text description with a mixture of Thai and English words and predicted symptom groups. We could say that we trained a combined model, and thus the evaluation was performed together.

We have revised the content on Page 7 Line 215 – 217 and Page 8 Lines 233 – 234.

[R2.9]

- “Model development”: While I appreciate the detail in this section, I think that there is actually too much information in the subsections for each model. In an already-complex paper, adding the extreme specifics of how each model works may be lost on readers from general audiences. Please consider moving portions of this section to a Methods supplement so that readers who are interested can read more, but make sure to keep enough in the main paper that it can be understood.

Answer

--------

Thank you very much for your suggestion. We kindly agreed with you. We have moved the mathematical descriptions of NB-SVM to the supporting information: "S1 Appendix: Natural Language Processing Models". We have revised the description of the other methods to contain only the implementation details that are of important for understanding of the manuscript. 

We have revised the content on Page 10 Line 278 – 287 and Page 10 – 11 Lines 312 – 323:

[R2.10]

- “Evaluation – Performance metrics”: Consider moving the mathematical descriptions of each of the different metrics to a Methods supplement, as a general audience likely will not understand this and get lost in the details.

Answer

--------

We have moved the mathematical descriptions of each of the different metrics into “S2 Appendix: Evaluation Metrices”

[R2.11]

Results

- Lines 430 – 432: This information could go in the “Evaluation” section of the methods instead of the Results.

Answer

--------

As the reviewer suggested, we have moved Lines 430 – 432 to “S2 Appendix: Evaluation Metrices”

[R2.12]

- Lines 441 – 443: I am confused on why Krippendorff’s alpha was used in this study if you are already assuming that the labels provided by the pharmacists are the gold standard.

Answer

--------

To illustrate how this algorithm can be integrated into the clinic, we selected the algorithm with the best performance, i.e., the ensemble model. We then prospectively tested the best-performing algorithm in a simulated EHR environment in which the algorithm suggests symptom groups from the free-text description. Pharmacists can choose to use the symptom terms suggested by our algorithm and can add or remove relevant symptom terms. This could be seen as augmented intelligence. Krippendorff's alpha is a metric we choose to show how our algorithm can help pharmacists code symptoms.

Krippendorff's alpha offers a different perspective on statistical measure compared to standard metrics for multi-label classification. 

[R2.13]

Discussion

- “Bilingual representation”: Because there were no results presented specifically for a bilingual model (or, if they were, they were done in a way that was not clear), this section seems more speculative than substantive. Please clarify in the methods how the bilingual component of the models were assessed, then describe this outcome in the Results. That way, this portion of the discussion will make more sense.

Answer

--------

The term “Bilingual representation” could be misleading as it may confer different meaning than what we intended to tell the readers. We simply wanted to say that our model can work in clinical documents that contains a mixture of Thai and English words. We agreed that it is difficult to perform assessment on that context. We have removed the name of the subsection and move the content to Page 14 – 15 Lines 442 – 451.

---

## [Decision Letter · Decision Letter 1]

4 Apr 2022

PONE-D-21-35704R1Multi-label classification of symptom terms from free-text bilingual adverse drug reaction reports using natural language processingPLOS ONE

Dear Dr. Chaichulee,

Thank you for submitting your manuscript to PLOS ONE. After careful consideration, we feel that it has merit but does not fully meet PLOS ONE’s publication criteria as it currently stands. Therefore, we invite you to submit a revised version of the manuscript that addresses the points raised during the review process.

We look forward to receiving your revised manuscript.

Kind regards,

Junaid Rashid, Ph.D

Academic Editor

PLOS ONE

Journal Requirements:

Please review your reference list to ensure that it is complete and correct. If you have cited papers that have been retracted, please include the rationale for doing so in the manuscript text, or remove these references and replace them with relevant current references. Any changes to the reference list should be mentioned in the rebuttal letter that accompanies your revised manuscript. If you need to cite a retracted article, indicate the article’s retracted status in the References list and also include a citation and full reference for the retraction notic

Reviewers' comments:

Reviewer's Responses to Questions

**Comments to the Author**

1. If the authors have adequately addressed your comments raised in a previous round of review and you feel that this manuscript is now acceptable for publication, you may indicate that here to bypass the “Comments to the Author” section, enter your conflict of interest statement in the “Confidential to Editor” section, and submit your "Accept" recommendation.

Reviewer #2: All comments have been addressed

Reviewer #3: (No Response)

2. Is the manuscript technically sound, and do the data support the conclusions?

Reviewer #2: Yes

Reviewer #3: Partly

3. Has the statistical analysis been performed appropriately and rigorously? 

Reviewer #2: Yes

Reviewer #3: N/A

4. Have the authors made all data underlying the findings in their manuscript fully available?

Reviewer #2: No

Reviewer #3: No

5. Is the manuscript presented in an intelligible fashion and written in standard English?

Reviewer #2: Yes

Reviewer #3: Yes

6. Review Comments to the Author

Reviewer #2: (No Response)

Reviewer #3: In this work authors evaluated different NLP algorithms that can encode unstructured ADRs stored in EHRs into institutional symptom terms.

Authors made efforts to revise the first draft of the manuscripts by addressing previous reviewers' comments. However, I have a few concerns on this paper and in my opinion, the manuscript is not in a state to be published. See my comments below

1) The approach leverages existing well known techniques together to solve an existing problem. It is not clear the key technical contribution of the proposed study. No novelty

2) There are many recent studies already published which are using the same idea even with more sophisticated ways of learning text representation. All recent studies are missing in the literature review.

3) Authors should refer to the state-of-the-art methods in Biomedical NLP (bioNLP) (e.g BioBERT and current SOTA BioALBERT). There are many studies which shows that using Biomedical (domain-specific) language models works better than language models trained on general corpus (such as Wikipedia etc). Authors should compare their results and discuss that domain-specific methods

BioALBERT: A Simple and Effective Pre-trained Language Model for Biomedical Named Entity Recognition

Benchmarking for Biomedical Natural Language Processing Tasks with a Domain Specific ALBERT

biobert: a pre-trained biomedical language representation model for biomedical text mining

4) The case for the paper is weak. The authors do provide a review of the relevant works however the relevant works are flatly discussed without properly highlighting their weaknesses and establishing the research gaps

5) Some experiment methods need more explanation

6) Finally, various typographical and grammatical errors must be rectified.

I would recommend that the authors look through more recent publications on this problem. Establishing novelty of approach over other published work would benefit their work, as well as the manuscript

7. PLOS authors have the option to publish the peer review history of their article (what does this mean?). If published, this will include your full peer review and any attached files.

Reviewer #2: No

Reviewer #3: No

---

## [Author Response · Author response to Decision Letter 1]

17 May 2022

Dear Academic Editor and Reviewers,

Thank you for your very thorough review of the manuscript. We greatly appreciate the reviewers' constructive comments, as well as the Academic Editor's decision to allow us to revise and resubmit unclear or overlooked points in our manuscript. We tried to maintain our original motivation and framework for developing the manuscript while correcting and improving the details and unclear statements based on the comments given. We have addressed all comments as described below. Our responses are as follows and are highlighted in yellow in the revised manuscript. We believe we have greatly improved the article and more clearly explained our findings.

Best Regards,

Sitthichok

Sitthichok Chaichulee DPhil

Department of Biomedical Sciences and Biomedical Engineering

Faculty of Medicine, Prince of Songkla University 90110, Thailand. 

Reviewer's Responses to Questions

Comments to the Author

1. If the authors have adequately addressed your comments raised in a previous round of review and you feel that this manuscript is now acceptable for publication, you may indicate that here to bypass the “Comments to the Author” section, enter your conflict of interest statement in the “Confidential to Editor” section, and submit your "Accept" recommendation.

Reviewer #2: All comments have been addressed

Reviewer #3: (No Response)

2. Is the manuscript technically sound, and do the data support the conclusions?

Reviewer #2: Yes

Reviewer #3: Partly

3. Has the statistical analysis been performed appropriately and rigorously?

Reviewer #2: Yes

Reviewer #3: N/A

4. Have the authors made all data underlying the findings in their manuscript fully available?

Reviewer #2: No

Reviewer #3: No

5. Is the manuscript presented in an intelligible fashion and written in standard English?

Reviewer #2: Yes

Reviewer #3: Yes

6. Review Comments to the Author

Reviewer #2: (No Response)

Reviewer #3:

In this work authors evaluated different NLP algorithms that can encode unstructured ADRs stored in EHRs into institutional symptom terms.

Authors made efforts to revise the first draft of the manuscripts by addressing previous reviewers' comments. However, I have a few concerns on this paper and in my opinion, the manuscript is not in a state to be published. See my comments below

Answer

Thank you for your very thorough review of the manuscript. We appreciate your comments very much. We have improved the article in many aspects and explained our contributions more clearly.

1) The approach leverages existing well-known techniques together to solve an existing problem. It is not clear the key technical contribution of the proposed study. No novelty.

Answer

--------

Our goal is to develop an approach that could help pharmacists to code symptoms. The problem of managing allergy data varies from institution to institution. Previous studies relied on rule-based techniques and the extensive use of data dictionaries. In contexts other than English, this process requires the creation of domain-specific data dictionaries, which is a very time consuming process. In the recent study on Russian ADRs, Lenivtceva and Kopanitsa (2021), developed their own rules and dictionaries which should be reviewed and updated over time. ML/DL offers an approach that does not require data dictionaries, but instead requires lots of data samples for training.

Our novelty relies on the investigation on the applicability of NLP in a domain-specific context, i.e., drug allergy, where there are still gaps in the development of such algorithms, especially in an environment where English is not the native language, as we pointed out in the Introduction section. In our drug allergy context, we generally explain clinical symptoms using free-text descriptions that are a mixture of Thai and English words. 

We kindly agree that we have chosen the standard conventional approaches, which are really mature for such implementation. We did, however, provide a thorough comparison of various algorithms, from simple NB-SVM to LSTM-based ULMFiT and newer transformer-based BERT models. We tried two multilingual BERT models: mBERT and XLM-ROBERTa, one bilingual model: WangchanBERTa and our AllergyROBERTa (pre-trained from scratch on our corpus). Our research reveals that XLM-RoBERTa, which was pre-trained on a very large multilingual general domain corpus, performs well in a clinical setting where a pre-trained LM is not available or data for training a new LM is missing.

We plan to explore NLP applications in other clinical contexts, such as inpatient coding of ICD-10 from discharge summary and clinical phenotype, which indeed require a more complex representation of clinical data.

2) There are many recent studies already published which are using the same idea even with more sophisticated ways of learning text representation. All recent studies are missing in the literature review.

Answer

--------

We have expanded the literature review of previous studies to provide more details about the studies in the Introduction section, Pages 4–5 Lines 144–173.

“Recently, there have been a few studies on the topics [5, 16–20]. Wagholikar et al. [16] developed a rule-based method using the SNOMED-CT (Systematized Nomenclature of Medicine – Clinical Terms) ontology for categorizing free-text chief complaints into symptom groups. The development of such algorithms was challenging due the different ways in which medical narratives were recorded by clinical staff. To deal with variations in clinical notes, Epstein et al. [5] developed an allergy-matching method that incorporates RxNorm, abbreviation, and misspelling terms to identify medication and food allergies from unstructured allergy records. Despite the use of extensive look-up tables, some free-text entries in the test set were not matched by the algorithm. Goss et al. [17] conducted a similar study with a larger set of terms, including SNOMED-CT and ICD-10 (International Classification of Diseases – 10th Revision) terms. Due to the presence of non-standard terms in free-text entries, algorithmic performances still varied. Similarly, Jackson et al. [18] developed SVM models to specifically capture key symptoms of severe mental illness from free-text discharge summaries using disease symptomatology concepts. Some symptoms were missed due to non-standard language usage. In languages other than English, Lenivtceva et al. [19] developed an extensive rule-based model for coding Russian free-text medical records into allergy identification categories that complies the HL7 FHIR (Health Level 7 – Fast Healthcare Interoperability Resources) standard. The method still suffered from lack of generalizability due to the availability of data dictionaries in Russian. Recently, Leiter et al. [20] using deep learning to identify symptoms in patients with congestive heart failure to assess the response of cardiac resynchronization therapy using an expert-labeled set of free-text notes without the representation of medical concepts. The algorithm was, however, inconsistently performed with typographical errors and phrasings that were absent from the training set. The problem could alleviate with a larger set of training samples. Some of the studies presented thus far required the modeling of medical concepts or the extensive use of data dictionaries. Most of the studies were conducted in a monolingual setting. However, in settings where English is not the official language, EHRs are often documented in another language or even a mixture of multiple languages.”

3) Authors should refer to the state-of-the-art methods in Biomedical NLP (bioNLP) (e.g BioBERT and current SOTA BioALBERT). There are many studies which shows that using Biomedical (domain-specific) language models works better than language models trained on general corpus (such as Wikipedia etc). Authors should compare their results and discuss that domain-specific methods

- BioALBERT: A Simple and Effective Pre-trained Language Model for Biomedical Named Entity Recognition

- Benchmarking for Biomedical Natural Language Processing Tasks with a Domain Specific ALBERT

- biobert: a pre-trained biomedical language representation model for biomedical text mining

Answer

--------

We intended to implement current SOTA LMs where available. Unfortunately, all publicly available biomedical and clinical LMs were developed based on an English-only corpus. They cannot be finetuned with our data, which contain a mixture of Thai and English words. For this reason, we chose to use multilingual LMs. Therefore, it is not possible to compare our results with those of biomedical and clinical LMs.

We have added the literature review of biomedical and clinical LMs on Page 4 Lines 114–127.

“Several LMs have been pre-trained on biomedical and clinical corpora. BioBERT was pre-trained over 18 billion words from PubMed abstracts and PubMed Central full-text articles [13]. ClinicalBERT was pre-trained on 2 million de-identified clinical notes from the Beth Israel Deaconess Medical Center [14]. BioALBERT [15] was pre-trained on the datasets used by BioBERT and ClinicalBERT, so the model learns language representation from both biomedical and clinical domain. All three models were reported to outperform the state of the art models pre-trained on general-domain data on most downstream tasks. However, they were all pre-trained with a monolingual English corpus. For the use of BERT models on clinical data, it was recommended to pre-train the model on a private dataset at the practitioner's institution for best results, as clinical notes may vary depending on the clinical setting [14].”

We have expanded the Discussion Section on this issue on Page 16 Lines 510–515.

“Although many studies have shown that biomedical and clinical LMs (such as BioBERT [13], ClinicalBERT [14], and BioALBERT [15]) performed better on downstream clinical tasks than general-domain LMs, all available domain-specific LMs have been pre-trained with a monolingual English corpus only. It is not possible to finetune these LMs with our institutional non-English dataset and compare their performances.”

4) The case for the paper is weak. The authors do provide a review of the relevant works however the relevant works are flatly discussed without properly highlighting their weaknesses and establishing the research gaps.

Answer

--------

We have rewritten the Discussion section Page 16–17 Lines 533–575 to highlight the weaknesses of the methods used by the previous studies and establish the research gaps.

“The extraction of clinical symptoms from unstructured clinical texts has been investigated in many studies [5, 16–20] The standardized symptomatic terms resulted from these works have been used for epidemiological research, clinical systems improvement, and healthcare interoperability. It is, however, difficult to make direct comparison to other studies because each study used its own institutional dataset and had its own unique purpose.

Compared with previous studies that used rule-based techniques [5, 16, 17, 19], our techniques were developed without the use of data dictionaries or clinical concept modeling. In the rule-based studies, Wagholikar et al. [16] created a script for mapping various synonyms and acronyms to SNOMED CT concepts. The approach was accurate for specific symptoms such as chest pain but not for more complex groups such as stomach pain and trauma. Epstein et al. [5] implemented an allergy-matching algorithm with Transact-SQL that included seven lookup tables. The algorithm achieved an F1 score of 0.98. The authors pointed out that periodically checking the remaining entries not found by the algorithm and periodically adding terms to the lookup tables could help improve the algorithm. Goss et al. [17] compiled several lists of standard terminologies with a set of rules defined for encoding allergy concepts. The authors noted a need for the algorithm to consider a contextual understanding between allergens and symptoms for best results. Recently, the rule-based pipeline that developed by Lenivtceva et al. [29] employed various techniques to handle misspellings and abbreviations, implemented over 20 regular expression rules, and constructed over four dictionaries of over 2,675 terms to describe allergy categories in Russian. Thanks to the complicated and specialized approach, the authors reported very good performance with F1-Score of 0.90–0.96 for allergy categorization. Implementing the rule-based approach could be time-consuming and tedious, but it is not necessary to have a well-curated large dataset.

The ML/DL approach has been reported to perform well, comparable to or better than rule-based methods, but depends heavily on the quality of the data. In practice, the ML/DL approach requires a well-curated dataset. Classical ML algorithms, such as SVM, have been employed in Jackson et al. [18] that involved human annotators to label clinical documents to in order to create training corpora. The approach can extract symptoms from clinical text with a median F1 score of 0.88 across 46 symptoms. In Leiter et al. [20], a DL algorithm was developed to identify symptoms from clinical notes of patients with congestive heart failure. The algorithm had an F1 score of 0.72 which might be due to a small sample size of only 154 notes.

Our study benefits from the availability of data that have already been labeled by experts over time. Without the need to define explicit rules, the ML/DL approach can reduce the time and effort required to develop clinical NLP algorithms. In addition, with a large training corpus, the ML/DL approach can learn some degree of noise in the text data, e.g. abbreviations, misspellings, uncertainties, and variations in clinical notes. However, the ML/DL approach may perform poorly on certain underrepresented symptom groups because the algorithm learns the categorization of the text from the data rather than from explicit rules.”

5) Some experiment methods need more explanation

Answer

--------

We have revised some experimental methods as per your comments.

6) Finally, various typographical and grammatical errors must be rectified.

Answer

--------

We apologize for the typographical and grammatical errors in the manuscript. We have now corrected the problem.

I would recommend that the authors look through more recent publications on this problem. Establishing novelty of approach over other published work would benefit their work, as well as the manuscript.

---

## [Decision Letter · Decision Letter 2]

14 Jun 2022

Multi-label classification of symptom terms from free-text bilingual adverse drug reaction reports using natural language processing

PONE-D-21-35704R2

Dear Dr. Chaichulee,

We’re pleased to inform you that your manuscript has been judged scientifically suitable for publication and will be formally accepted for publication once it meets all outstanding technical requirements.

Kind regards,

Junaid Rashid, Ph.D

Academic Editor

PLOS ONE

Additional Editor Comments :

I believe the authors have addressed all reviewer comments, and the manuscript can be accepted.

Reviewers' comments:

Reviewer's Responses to Questions

**Comments to the Author**

1. If the authors have adequately addressed your comments raised in a previous round of review and you feel that this manuscript is now acceptable for publication, you may indicate that here to bypass the “Comments to the Author” section, enter your conflict of interest statement in the “Confidential to Editor” section, and submit your "Accept" recommendation.

Reviewer #3: All comments have been addressed

2. Is the manuscript technically sound, and do the data support the conclusions?

Reviewer #3: Partly

3. Has the statistical analysis been performed appropriately and rigorously? 

Reviewer #3: N/A

4. Have the authors made all data underlying the findings in their manuscript fully available?

Reviewer #3: (No Response)

5. Is the manuscript presented in an intelligible fashion and written in standard English?

Reviewer #3: (No Response)

6. Review Comments to the Author

Reviewer #3: Authors have addressed most of my comments. I have no more suggestions there I recommend to accept this paper for publication.

7. PLOS authors have the option to publish the peer review history of their article (what does this mean?). If published, this will include your full peer review and any attached files.

Reviewer #3: No

---

## [Editor Report · Acceptance letter]

7 Jul 2022

PONE-D-21-35704R2 

Multi-label classification of symptom terms from free-text bilingual adverse drug reaction reports using natural language processing 

Dear Dr. Chaichulee:

I'm pleased to inform you that your manuscript has been deemed suitable for publication in PLOS ONE. Congratulations! Your manuscript is now with our production department. 

Kind regards, 

on behalf of

Dr. Junaid Rashid 

Academic Editor

PLOS ONE